# MLL-AF4 cooperates with PAF1 and FACT to drive high-density enhancer interactions in leukemia

Nicholas T. Crump [1,2,10] ✉, Alastair L. Smith [1,10], Laura Godfrey[1], Ana M. Dopico-Fernandez [1], Nicholas Denny[1], Joe R. Harman [1], Joseph C. Hamley[1], Nicole E. Jackson [1], Catherine Chahrour [1], Simone Riva[1], Siobhan Rice[1], Jaehoon Kim [3], Venkatesha Basrur[4], Damian Fermin[4], Kojo Elenitoba-Johnson[5], Robert G. Roeder [6], C. David Allis [7,11], Irene Roberts[1,8], Anindita Roy [1,8], Huimin Geng[9], James O. J. Davies[1] & Thomas A. Milne [1] ✉

Aberrant enhancer activation is a key mechanism driving oncogene expression in many cancers. While much is known about the regulation of larger chromosome domains in eukaryotes, the details of enhancer-promoter interactions remain poorly understood. Recent work suggests co-activators like BRD4 and Mediator have little impact on enhancer-promoter interactions. In leukemias controlled by the MLL-AF4 fusion protein, we use the ultra-high resolution technique Micro-Capture-C (MCC) to show that MLL-AF4 binding promotes broad, high-density regions of enhancer-promoter interactions at a subset of key targets. These enhancers are enriched for transcription elongation factors like PAF1C and FACT, and the loss of these factors abolishes enhancer-promoter contact. This work not only provides an additional model for how MLL-AF4 is able to drive high levels of transcription at key genes in leukemia but also suggests a more general model linking enhancer-promoter crosstalk and transcription elongation.

There is a growing understanding that alterations in the three-dimensional structure of the genome (the so-called 3D genome) can have a profound impact on gene expression[1–3]. Much is now known about the factors that govern higher-order 3D genome structure, but the mechanistic details that control enhancer-promoter crosstalk remain poorly understood[3].

Enhancers are key regulatory elements that contribute to gene expression. Aberrant enhancer activity is a major factor in many cancers, often involving specific DNA mutations or even large-scale DNA rearrangements[4,5]. Epigenetic changes in cancer are also increasingly recognized as important drivers of gene expression changes[6]. Active enhancers commonly display characteristics that include an open

[1]MRC Molecular Haematology Unit, MRC Weatherall Institute of Molecular Medicine, Radcliffe Department of Medicine, University of Oxford, Oxford OX3 9DS, UK. [2]Hugh and Josseline Langmuir Centre for Myeloma Research, Centre for Haematology, Department of Immunology and Inflammation, Imperial College London, London W12 0NN, UK. [3]Department of Biological Sciences, Korea Advanced Institute of Science and Technology, Daejeon 34141, South Korea. [4]Department of Pathology, University of Michigan, Ann Arbor, Michigan, USA. [5]Department of Pathology and Laboratory Medicine, Perelman School of Medicine, University of Pennsylvania, Philadelphia, Pennsylvania, USA. [6]Laboratory of Biochemistry and Molecular Biology, The Rockefeller University, New York, NY 10065, USA. [7]Laboratory of Chromatin Biology and Epigenetics, The Rockefeller University, New York, NY 10065, USA. [8]Department of Paediatrics, University of Oxford, Oxford OX3 9DU, UK. [9]Department of Laboratory Medicine, University of California, San Francisco, San Francisco, California, USA. [10]These authors contributed equally: Nicholas T. Crump, Alastair L. Smith. [11]Deceased: C. David Allis. ✉e-mail: n.crump@imperial.ac.uk; thomas.milne@imm.ox.ac.uk

chromatin conformation, close spatial proximity to the target promoter[3,7,8], bidirectional transcription of unstable enhancer RNA (eRNA)[9], and post-translational histone modifications such as histone H3 lysine-4 monomethylation (H3K4me1) and H3 lysine-27 acetylation (H3K27ac)[10].

Exactly how enhancers control gene expression remains unknown, although they are thought to function in part by acting as docking sites for transcription factors, which then activate appropriate target genes over long distances[1]. Enhancers have been proposed to drive transcription at both initiation[11,12] as well as promoter-proximal pause release[13,14]. Live imaging has suggested that strong enhancers impact the transcription cycle by increasing burst frequency rather than burst size[15,16], arguing that one of the main roles of enhancers is to increase the probability of successful transcription events.

The factors that initiate enhancer-promoter crosstalk may be different from those that maintain it. For instance, during embryonic stem cell differentiation, pre-binding of the Polycomb Repressive Complex (PRC) is required for initial enhancer-promoter contacts[17,18], but this does not seem to be necessary for inducing gene expression during differentiation[18]. Transcription factor binding has also been thought to be essential for the initiation of enhancer-promoter crosstalk, but this has not been shown directly. Co-activators such as BRD4 and Mediator have been proposed to bridge the gap between transcription factor binding and enhancer-promoter contact[19–25], although recent work from us and others suggests that these co-activators are not necessary for the maintenance of enhancer-promoter proximity or at least only have a subtle role[26,27]. However, we identified a subset of enhancers, H3K79me2/3-marked Enhancer Elements (KEEs), which are highly dependent on this methylation for the maintenance of gene expression and enhancer-promoter interactions[28]. What is not clear is whether other components of the transcriptional machinery might have similar crucial impacts on maintaining enhancer activity and whether they are necessary for the maintenance of enhancer-promoter crosstalk.

Rearrangements of the *Mixed Lineage Leukemia* (*MLL*, also known as *KMT2A*) gene (*MLL*r) cause aggressive, poor prognosis acute lymphoblastic (ALL) and acute myeloid (AML) leukemias in both children and adults[29–31] and are associated with very few cooperating mutations[32–36]. Despite the relatively simple genetic landscape, *MLL*r leukemias exhibit a large number of epigenetic and transcriptional changes, suggesting that the MLL fusion protein (MLL-FP) drives oncogenesis via transcriptional reprogramming and therefore provides a good model to study aberrant activation of enhancers.

MLL-AF4, along with other MLL-FPs, is thought to drive leukemia by binding at the promoters of key oncogenes and upregulating their expression by the recruitment of a complex of elongation-associated factors, including the RNA Polymerase (RNAP) II-Associated Factor complex (PAF1C), Eleven Nineteen Leukemia (ENL, also known as MLLT1), AF9, DOT1L, AFF4 and P-TEFb[37–39]. In many cases, MLL-FP binding spreads from the promoter into the gene body, associated with elevated levels of elongation factors and upregulated transcription[40]. Many target genes are also regulated by strong enhancers, and we and others have identified MLL-AF4 binding at intragenic and intergenic enhancers[41–43], but the significance of this behavior and whether it is dependent on the same factors associated with MLL-AF4 at gene promoters has not been established.

In this study, we seek to understand the role of MLL-AF4 in enhancer function as a model to investigate mechanisms of aberrant enhancer activation in cancer. Using Micro-Capture-C (MCC), a highly sensitive 3C method[8], we show that MLL-AF4 binding is a major driver of aberrant enhancer activation, characterized by a high density of interactions between enhancers and promoters of key genes that drive leukemogenesis. Many of the proteins colocalizing at MLL-AF4-bound enhancers are the same elongation factors that associate with MLL-AF4 at target genes, including ENL and PAF1C. We identify here the histone chaperone Facilitates Chromatin Transcription (FACT) as an MLL-AF4 complex component found at enhancers, which is typically reported to be associated with transcription elongation. Chemical degradation of either PAF1 or the FACT component SSRP1 results in a drastic loss of enhancer-promoter interactions at many MLL-AF4-bound and -unbound enhancers, indicating that these factors are generally required for enhancer function. Together, this reveals an unexpected mechanism of oncogenicity for the MLL-AF4 fusion protein in maintaining enhancer-promoter contacts and argues that highly enriched transcription elongation activity can drive high levels of enhancer activity.

## Results

### MLL-AF4 binds at oncogene enhancers

To define the binding sites of MLL-AF4 and other MLL-AF4 complex members at high resolution in primary samples, we adapted high-throughput ChIPmentation[44], optimizing it for analysis of more weakly-associated non-histone chromatin proteins, an approach we call Transcription factor-OPtimized ChIPmentation (TOPmentation, Supplementary Fig. 1a). The TOPmentation method generated highly comparable data to that produced by standard ChIP-seq using ~100-fold fewer cells (approx. 100,000 cells), allowing the chromatin environment to be delineated even in patient samples where few primary leukemia cells are available (Supplementary Fig. 1b, c).

We profiled four primary MLL-AF4 ALL patient samples (three males and one female), including three childhood (1–18 years old; chALL#1, chALL#2 and chALL#3) and one infant (≤1 year old; iALL#2), combining gene expression and chromatin accessibility data with TOPmentation and ChIP-seq for H3K4me1, H3K4me3, H3K27ac, H3K27me3, H3K79me2 and the N-terminus of MLL-AF4 (Supplementary Table 1). In addition, we analyzed MLL-AF4 binding in a published iALL patient primograft (iALL#1, male ≤1 year old)[40,45]. Strikingly, we observed a very high concordance in MLL-AF4 promoter binding between the patient samples (using an antibody recognizing the N-terminus of MLL as a proxy for fusion protein binding). Approximately 60% of MLL-AF4 bound genes (6021) were common between all patients, and very few promoters (≤184) were bound in only one sample (Supplementary Fig. 1d). Consistent with its role in promoting gene expression, approximately 80% of bound genes were transcriptionally active (Supplementary Fig. 1e), and MLL binding was highly correlated with H3K27ac levels at promoters (Fig. 1a).

While many MLL-AF4 peaks were present within 1 kb of a TSS (i.e., promoter bound), a large proportion was found more than 10 kb from the nearest TSS (Fig. 1b). We observed a similar distribution in a range of *MLL*r ALL and AML cell lines (Supplementary Fig. 1f), indicating that non-promoter binding is a widespread property of MLL-FPs. The majority of these distal MLL-FP binding sites overlapped with peaks of H3K27ac, implying binding at active enhancers (Supplementary Fig. 1g). Thus, MLL-AF4 enhancer binding is suggestive of an additional mechanism for regulating gene expression[41–43].

Strikingly, we found MLL-AF4-bound enhancers in proximity to a number of key oncogenes, including *FLT3*, *MYC*, *CDK6* (Fig. 1c, Supplementary Fig. 1h) and *PROM1*[41], suggesting a potentially important role for enhancer activation in driving and/or maintaining leukemogenesis. Notably, while many enhancers were associated with the spreading of MLL-AF4 into the gene body from the promoter[40,41], we also identified MLL-AF4 binding at intergenic enhancers, for example, the *MYC* enhancer which is more than 1.5 Mb away from the promoter (Supplementary Fig. 1h, i). MLL-AF4 enhancer binding may therefore play an important role in determining the biology of these leukemias.

The impact of MLL-AF4 on enhancer function has not been explored in detail, especially in patients. Using our patient data, we identified putative enhancers by intersecting non-promoter open-chromatin sites with H3K27ac peaks. Strikingly, a substantial proportion of MLL-AF4 binding sites overlapped with putative enhancers

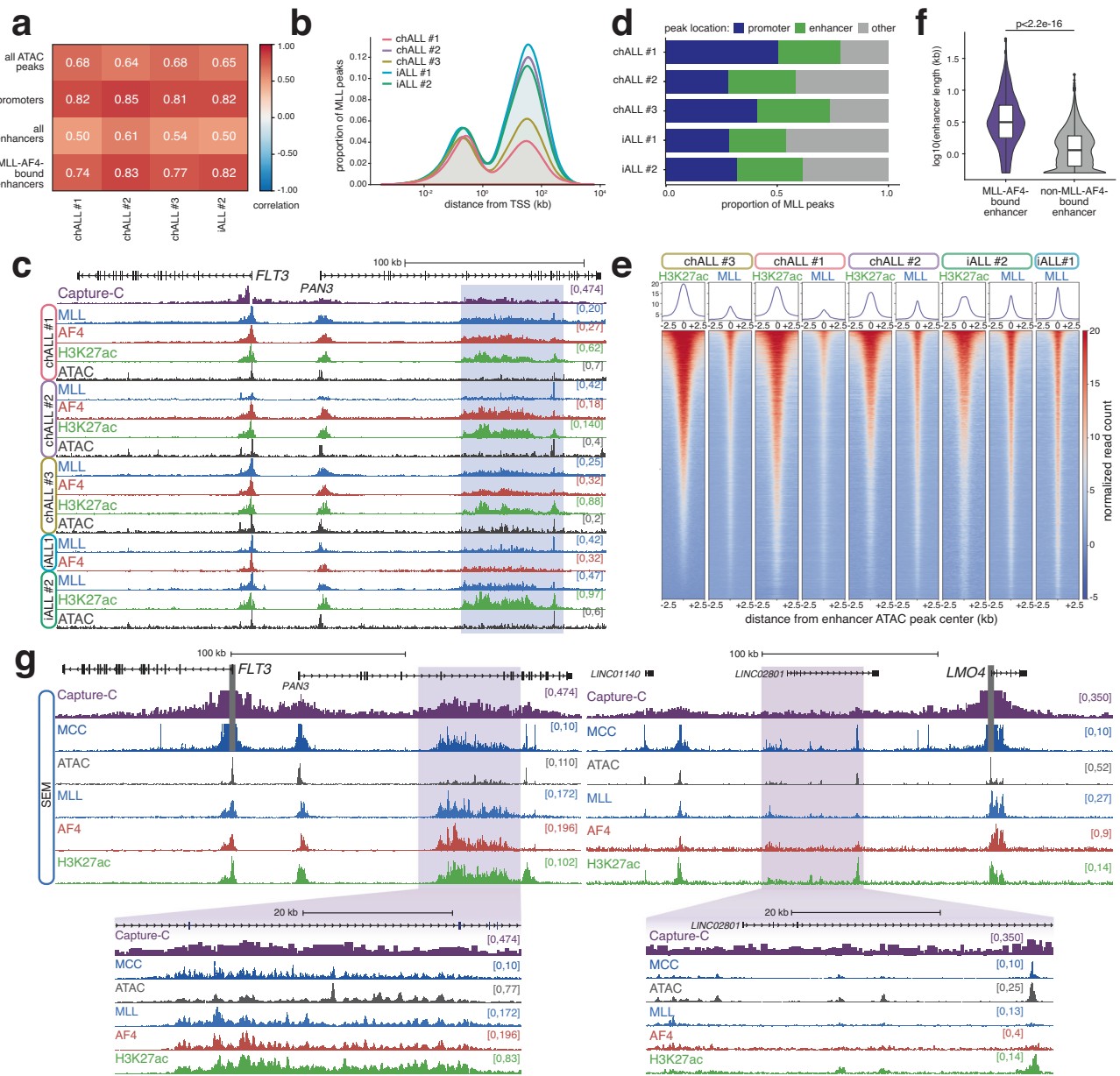

**Fig. 1 | MLL-AF4 binding at enhancers is a common feature of cell lines and primary material. a** Correlation of H3K27ac and MLL ChIP-seq/TOPmentation signal at all ATAC peaks, promoter ATAC peaks, all enhancers and MLL-AF4-bound enhancers for the indicated patient samples. **b** Distribution of MLL peaks in five MLL-AF4 patient samples relative to the nearest TSS. **c** ChIP-seq/TOPmentation for MLL, AF4 and H3K27ac, and ATAC-seq at the *FLT3*/*PAN3* locus in the indicated patient samples. Capture-C from the *FLT3* TSS is shown for SEM cells. The MLL-AF4-bound enhancer within *PAN3* is highlighted in blue. **d** Proportion of MLL peaks associated with promoters and enhancers in each patient sample. **e** Heatmap of H3K27ac ChIP-seq and MLL ChIP-seq/TOPmentation signal in the indicated patient samples at enhancer ATAC-seq peaks. **f** Distribution of the length of enhancers bound (*n* = 807) or not bound (*n* = 8948) by MLL-AF4 in SEM cells. *p*-value indicates the statistical significance of the difference in enhancer length (two-sided Wilcoxon rank sum test), *p* < 2.2 × 10⁻¹⁶. Midline shows median, with upper and lower hinges showing the 25th and 75th percentile, respectively. Upper and lower hinges extend to the largest and smallest datapoints within 1.5 times the interquartile range of either hinge. **g** Capture-C, Micro-Capture-C (MCC), ATAC-seq and ChIP-seq for MLL, AF4 and H3K27ac at the *FLT3* and *LMO4* loci in SEM cells. Enhancer regions are highlighted in purple. Capture-C and MCC traces scaled to emphasize distal interactions.

(Fig. 1d). MLL-AF4 binding at these enhancers was strongly correlated between patients (Supplementary Fig. 1j) with 2550 enhancers bound by MLL-AF4 in at least three of the four patients (Supplementary Fig. 2a, Supplementary Data 1). Within each patient, we also observed a high degree of correlation between the levels of H3K27ac and MLL-AF4 binding at MLL-AF4 bound enhancers (Fig. 1a, e), again indicating that MLL-AF4 is involved in enhancer activity in addition to promoter activity in primary ALL cells. To assess a possible functional role for these enhancers, we used a nearest-neighbor approach to assign putative enhancers to genes[10,28]. In all four patients, genes associated

with an MLL-AF4-bound enhancer were more highly expressed than other genes (Supplementary Fig. 2b), implying a direct role for these enhancers in upregulating transcription.

We asked whether the 2550 enhancers bound by MLL-AF4 were unique to MLL-AF4 ALL by comparing H3K27ac levels at these enhancers, as a proxy for activity, in cell lines representing different ALL subtypes (Supplementary Fig. 2c and Supplementary Data 1)[46]. As expected, *MLL*r cell lines clustered with MLL-AF4 patients and showed unique activity at a subset of MLL-AF4-bound enhancers (cluster 4), for example, at the *FLT3* enhancer (Supplementary Fig. 2d, left). Genes

that are associated with these enhancers include canonical MLL-AF4 targets, such as *MEIS1, FLT3, PROM1, HOXA7, CCNA, CPEB2, RUNX2, ARID1B, MBNL1* and *JMJD1C*[28,40,41,47] (Supplementary Data 1). In addition to the MLL-AF4-unique enhancers, some enhancer loci appeared active across a broader subset of ALL cell lines (cluster 3), and others appeared to be more highly active in non-*MLL*r cell lines (clusters 1 and 2), for example, at *TNFRSF14* (Supplementary Fig. 2d, right). Thus, while a subset of MLL-AF4-bound enhancers are unique to *MLL*r leukemia, many others are likely to be activated by additional MLL-AF4-independent mechanisms in other ALLs.

To further examine the specificity of the MLL-AF4-unique enhancers, we used a published patient RNA-seq dataset[36] to compare the expression of enhancer-associated genes in *MLL*r and *MLL*wt leukemia. Genes associated with the MLL-AF4-unique enhancers (cluster 4) were more highly expressed in *MLL*r ALL (Supplementary Fig. 2e). In contrast, enhancers active in other ALL subtypes showed a similar level of expression in *MLL*r and *MLL*wt ALL. Using a complementary approach, we matched our 2550 MLL-AF4 enhancer gene set to 881 genes from a published ALL patient microarray dataset[48] and ranked them based on their ability to distinguish *MLL*r patient samples from other ALL subtypes. More than half of the top 50 genes were associated with the MLL-AF4-unique enhancers in cluster 4 (Supplementary Fig. 2f). Finally, using four different published patient datasets, we found that genes associated with MLL-AF4-unique enhancers were significantly overexpressed in *MLL*r patient samples compared to other ALL subtypes or normal preB cells (Supplementary Fig. 2g; ECOG E2993[48] 84 genes up vs 36 down, $p = 1.4 \times 10^{-5}$; COG P9906[49] 114 genes up vs 16 down, $p < 1 \times 10^{-10}$; St. Jude 2003[50] 83 genes up vs 28 down, $p = 1.7 \times 10^{-7}$; St. Jude 2013[51] 70 genes up vs 14 down, $p = 4.1 \times 10^{-10}$). Taken together, this suggests that MLL-AF4 binding to enhancers contributes to the unique gene expression pattern observed in MLL-AF4 leukemia.

## MLL-AF4 binding can be associated with aberrantly large enhancers that display high-density 3D interactions with promoters

In general, MLL-AF4-bound enhancers are longer than non-MLL-AF4-bound enhancers (Fig. 1f, Supplementary Fig. 3a), with some spanning over 50 kb, for example, at *FLT3, MYC* and *CDK6* (Fig. 1c, Supplementary Fig. 1h). A key feature of super-enhancers (SEs) is their extended length[52], so we called SEs in the MLL-AF4 cell line SEM to ask whether they overlapped with MLL-AF4-bound enhancers. Indeed, about half of MLL-AF4-bound enhancers were classified as SEs (Supplementary Fig. 3b), suggesting that MLL-AF4 associates with, and may facilitate, strong enhancer activity. Notably, even within SEs, MLL-AF4 binding was associated with increased enhancer length (Supplementary Fig. 3c).

Visual inspection of ATAC-seq data at key broad MLL-AF4-bound enhancers, for example, at the *FLT3, ARID1B, CDK6* and *MYC* enhancer loci (Fig. 1g, Supplementary Fig. 3d), revealed a high density of peaks of open chromatin, especially when compared to other enhancers, such as at *LMO4, IKZF3* and *SMAD3* (Fig. 1g, Supplementary Fig. 3e), suggesting a high frequency of protein binding. This increased accessibility appears to be associated with leukemogenesis, as, for example, the high density of ATAC-seq peaks observed at the *FLT3* enhancer in MLL-AF4 patients and cell lines is completely absent in normal cells, including those from the B lineage (Supplementary Fig. 2d)[53,54]. We explored whether MLL-AF4 associates with highly accessible enhancers genome-wide by comparing ATAC-seq signal enrichment at the 2550 MLL-AF4-bound enhancer set (Supplementary Data 1) with published ATAC-seq from normal hematopoietic cells[53]. While some of these enhancers displayed highly enriched signals, specifically in the MLL-AF4 patient blasts (Supplementary Fig. 3f; cluster 3), a large proportion also showed high levels of accessibility in common lymphoid progenitor (CLP) cells (clusters 4 and 5). Other MLL-AF4

enhancers, for example, at *TNFRSF14* (Supplementary Fig. 2d), were more active in non-B lineage cell types (clusters 1 and 2). The MLL-AF4 specific ATAC cluster (cluster 3) contained such canonical genes as *FLT3* (Supplementary Fig. 2d), *PROM1, RUNX2, ARID1B, MBNL1* and *JMJD1C* (Supplementary Data 1). Of the 300 MLL-AF4 specific enhancers in ATAC cluster 3, 124 (41%) were highly enriched for H3K27ac signal compared to other ALL samples (Supplementary Fig. 3g; i.e., are found in H3K27ac cluster 4, Supplementary Fig. 2c). In general, this suggests that MLL-AF4 binding is associated with novel enhancer activity at a subset of key target genes (for example at *FLT3* and *PROM1*). In other cases, MLL-AF4 binds to pre-existing enhancers and may contribute to maintaining their activity in leukemia.

A key feature of most enhancers is a high frequency of interactions with target gene promoters. To explore this at MLL-AF4-bound enhancers, we used the ultra-high resolution technique Micro-Capture-C (MCC)[8] to look at a subset of key oncogenes. Although MCC does not allow genome-wide interaction analysis, the high sequencing depth permits precise mapping of DNA-DNA interactions at single base pair resolution[8]. The MCC interaction profiles for *FLT3, ARID1B, CDK6* and *MYC* are markedly broad, showing extensive interactions aligning with the high density of ATAC-seq peaks (likely TF binding events)[8] (Fig. 1g, left, Supplementary Fig. 3d). These interactions also broadly correlate with MLL-AF4 binding, suggesting that regions densely bound by MLL-AF4 directly contact the promoter (Fig. 1g, Supplementary Fig. 3d). This does not appear to be a general feature of enhancers in SEM cells, as at genes associated with non-MLL-AF4 enhancers, such as *LMO4, IKZF1* and *SMAD3*, (Fig. 1g, right, Supplementary Fig. 3e), MCC reveals more punctate interaction loci. Since MCC is not a genome-wide technique, we were not able to verify what proportion of MLL-AF4 bound enhancers display these high-density interaction profiles. However, we conclude that, at a subset of highly active enhancers, MLL-AF4 binding is associated with a large-scale hub of contacts (spanning tens of kb) with target promoters to activate transcription.

## MLL-AF4 enhancer binding drives transcription of key oncogenes

Having established that MLL-AF4 binds to a subset of active enhancers at key target genes, we asked whether it was required for their function. We have previously demonstrated that MLL-AF4 knockdown in SEM cells results in the downregulation of a large number of genes[40]. We intersected SEM enhancers and MLL-AF4 peaks, identifying enhancers enriched (bound) or depleted (not bound) for MLL-AF4 (Fig. 2a), and linked these to genes using a nearest-neighbor approach[10,28]. Genes associated with an MLL-AF4-bound enhancer were more strongly downregulated following MLL-AF4 knockdown (Fig. 2b), even when accounting for the presence/absence of MLL-AF4 at the promoter (Supplementary Fig. 4a). A greater proportion of MLL-AF4 enhancer-associated genes were downregulated (Fig. 2b, Supplementary Fig. 4b), suggesting that MLL-AF4 binding at these enhancers upregulates gene expression.

MLL-AF4 enrichment at enhancers correlated with elevated H3K27ac and enhancer RNA (eRNA) transcription, suggesting that MLL-AF4 is associated with high levels of enhancer activity (Fig. 2c). Surprisingly, knockdown of MLL-AF4 (Supplementary Fig. 4c, d) resulted in a general reduction in enhancer acetylation (Fig. 2c, Supplementary Fig. 4e), including at key oncogene enhancers (Fig. 2d, Supplementary Fig. 4f). The effect was confirmed by ChIP-qPCR in both SEM and RS4;11 cells at common and cell line-specific enhancers (Fig. 2e, Supplementary Fig. 4d). In contrast, we observed a stronger reduction in eRNA transcription specifically from MLL-AF4-bound enhancers (Fig. 2c, right, Supplementary Fig. 4e), suggesting that this change is a direct consequence of loss of MLL-AF4 binding. Together, these results strongly indicate a functional role for MLL-AF4 in enhancer activity.

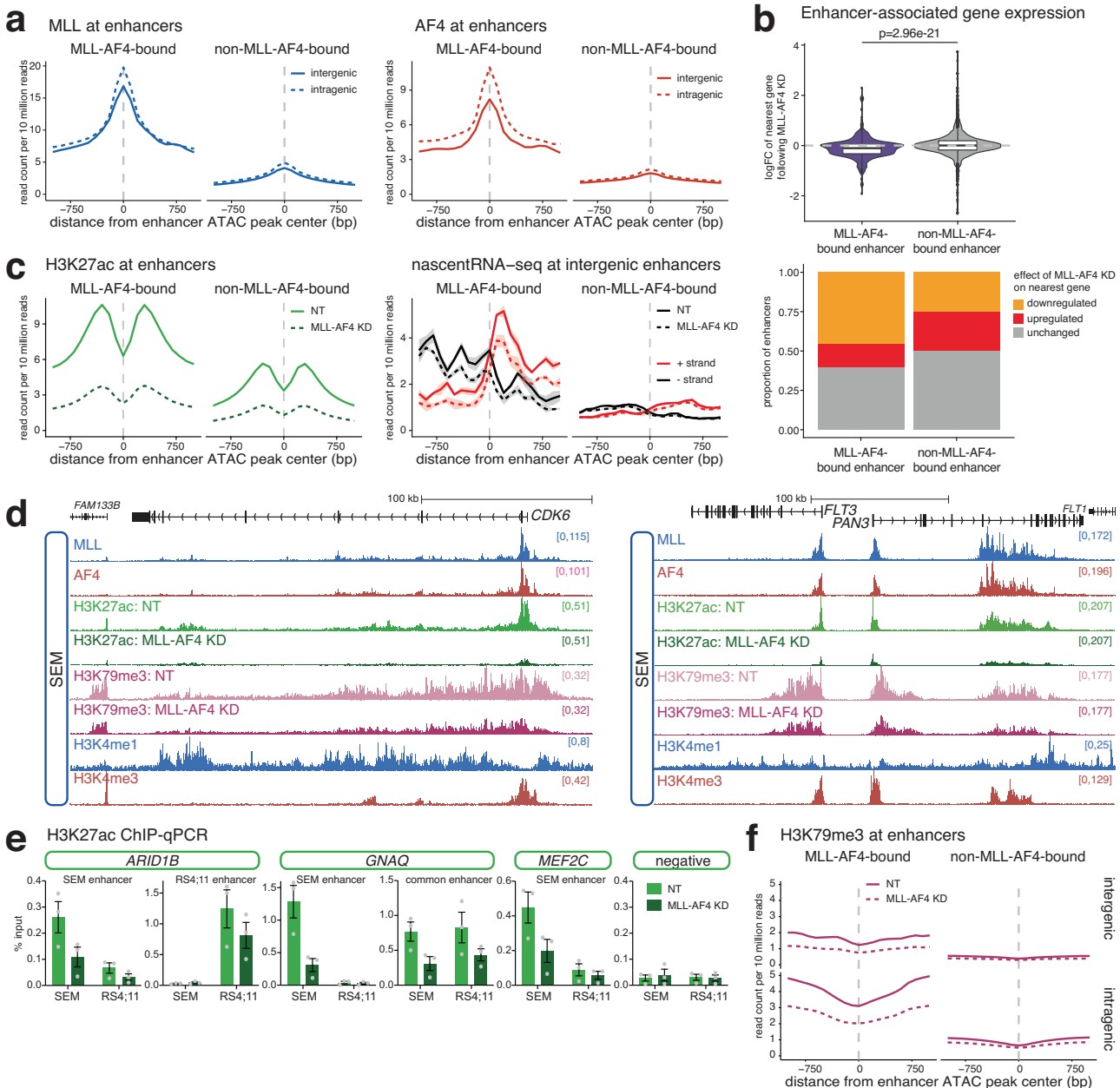

**Fig. 2 | MLL-AF4 binding is required for the maintenance of enhancer signatures. a** Mean distribution of MLL (*left*) and AF4 (*right*) at MLL-AF4-bound and non-MLL-AF4-bound intergenic (solid line) or intragenic (dashed line) enhancers in SEM cells. Plots are centered on ATAC-seq peaks found within enhancers. **b** *Upper*: Mean log-fold change in gene expression in SEM cells following 96 h MLL-AF4 knockdown for genes associated with an MLL-AF4-bound enhancer or genes associated with an enhancer not bound by MLL-AF4, *n* = 3 independent experiments. Statistical significance calculated using a two-sided Mann–Whitney U test, *p* = 2.96 × 10⁻²¹. Midline shows median, with upper and lower hinges showing the 25th and 75th percentile, respectively. Upper and lower hinges extend to the largest and smallest datapoints within 1.5 times the interquartile range of either hinge. *Lower*: Proportion of enhancers associated with genes displaying each transcriptional response to MLL-AF4 knockdown. **c** Mean distribution of H3K27ac (*left*) and strand-specific nascent RNA-seq (enhancer RNA; *right*) levels at MLL-AF4-bound and non-MLL-AF4-bound enhancers, in SEM cells under control (NT) and 96 h MLL-AF4 knockdown conditions. Lines represent mean, shading represents ± SEM, *n* = 3 independent experiments for eRNA. **d** Reference-normalized ChIP-seq for H3K27ac and H3K79me3 at *CDK6* and *FLT3* in SEM cells under control (NT) and 96 h MLL-AF4 knockdown conditions. ChIP-seq for MLL, AF4, H3K4me1 and H3K4me3 is shown for context. **e** ChIP-qPCR for H3K27ac at the indicated enhancer regions in SEM and RS4;11 cells, under control (NT) and 96 h MLL-AF4 knockdown conditions. Data are represented as mean ± SEM, *n* = 3 independent experiments. Source data are provided as a Source Data file. **f** Mean distribution of H3K79me3 at MLL-AF4-bound and non-MLL-AF4-bound inter- and intragenic enhancers in SEM cells under control (NT) and MLL-AF4 knockdown conditions.

We have previously shown that H3K79me2/3 is required to maintain the activity of a subset of enhancers, termed H3K79me2/3-marked enhancer elements (KEEs)[28]. MLL-AF4 is associated with elevated H3K79me2/3[55,56], and both intragenic and intergenic enhancers bound by MLL-AF4 were enriched for H3K79me3 (Fig. 2f). Indeed, many KEEs are bound by MLL-AF4 (Supplementary Fig. 4g), and this

mark was depleted following MLL-AF4 knockdown (Fig. 2f, Supplementary Fig. 4e), suggesting that MLL-AF4 may be responsible for generating a subset of KEEs in SEM cells. For example, the MLL-AF4-bound KEEs within *CDK6*, *FLT3* and *ARID1B* showed a strong depletion of H3K27ac and H3K79me3 (Fig. 2d, Supplementary Fig. 4f), associated with downregulation of gene transcription (Supplementary Fig. 4h).

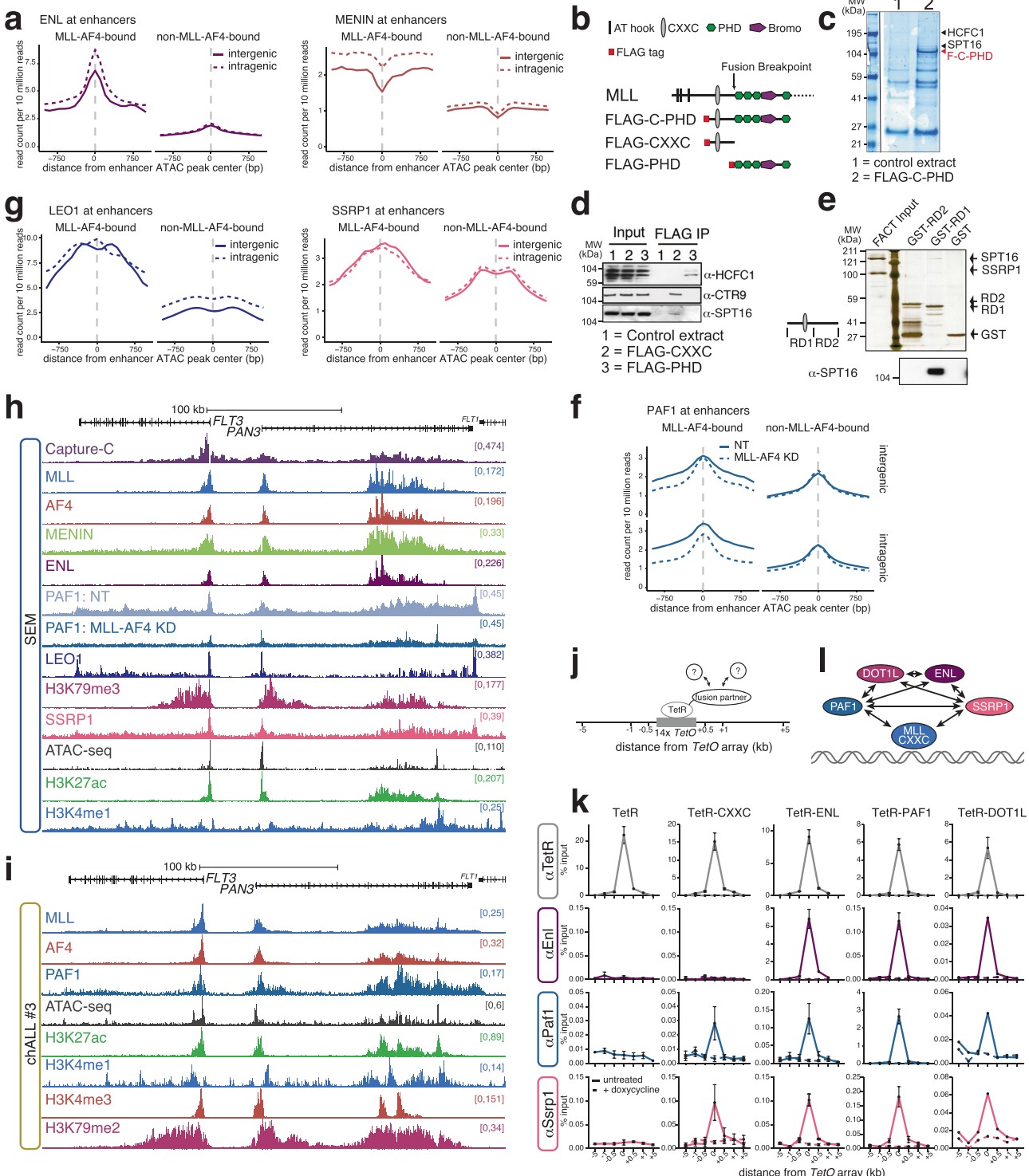

## MLL-AF4 drives high-density enhancer interactions by recruiting transcription elongation factors

MLL-AF4 is known to interact with a core set of factors at promoters, including DOT1L, MENIN, ENL and many other proteins[57–59]. We therefore asked whether MLL-AF4 recruits these components to enhancers to promote enhancer activity. The elevated levels of H3K79me3 at MLL-AF4 enhancers (Fig. 2f) suggested the assembly of a functional complex of factors, including DOT1L. In addition, MLL-AF4 enhancers were enriched for MENIN and ENL (Fig. 3a). Binding was observed at both intergenic and intragenic enhancers, indicating that

the proteins were not present as an indirect consequence of increased transcription within the gene body (Fig. 3a).

In addition to the MLL-AF4 core complex, we looked for additional transcriptional proteins that may be recruited to enhancers by MLL-AF4. Screening a MLL mass spectrometry dataset for additional co-activators, we identified the histone chaperone FACT, a dimer of SPT16 and SSRP1, as a potential MLL-interacting protein complex member (Fig. 3b, c, Supplementary Data 2). To validate the putative interaction between MLL-AF4 and FACT, we performed immunoprecipitation (Fig. 3d) and GST pulldown experiments (Fig. 3e), demonstrating that purified FACT interacts directly with the CXXC domain-

**Fig. 3 | MLL-AF4 binding recruits transcription elongation factors to enhancers. a** Mean distribution of ENL and MENIN at MLL-AF4-bound and non-MLL-AF4-bound intergenic (solid line) and intragenic (dashed line) enhancers in SEM cells. **b** Schematic of wild-type N-terminal MLL structure, showing the domains used for immunoprecipitation. **c** Colloidal Blue-stained gel of control HEK-293 nuclear extracts (1) or HEK-293 nuclear extracts expressing the FLAG-C-PHD construct (2), immunoprecipitated with anti-FLAG antibody. Gel lanes were sliced and subjected to mass spectrometry (see "Methods"). Regions where the proteins HCFC1 and FACT complex component SPT16 were identified are indicated by arrowheads. The red arrowhead indicates the position of the FLAG-C-PHD protein. Image represents a single replicate used for MS. Source data are provided as a Source Data file. **d** Immunoblots for HCFC1, CTR9 (PAF1C component) and SPT16 following anti-FLAG immunoprecipitation of HEK-293 cell lysates expressing the indicated FLAG-tagged MLL domains. Representative of three experiments. Source data are provided as a Source Data file. **e** Silver-stained gel after affinity purification of GST-tagged MLL RD1 and RD2 domains following incubation with purified SPT16 and SSRP1 (FACT). Lower panel shows immunoblot for SPT16. Representative of two experiments. Source data are provided as a Source Data file. **f** Mean distribution of PAF1 at MLL-AF4-bound and non-MLL-AF4-bound intragenic and intergenic enhancers in SEM cells under control (NT; solid line) and 96 h MLL-AF4 knockdown (dashed line) conditions. **g** Mean distribution of PAF1C component LEO1 and FACT component SSRP1 at MLL-AF4-bound and non-MLL-AF4-bound intergenic (solid line) and intragenic (dashed line) enhancers in SEM cells. **h** ChIP-seq, ATAC-seq and Capture-C at the *FLT3* locus in SEM cells. Reference-normalized ChIP-seq for PAF1 in SEM cells under control (NT) and 96 h MLL-AF4 knockdown conditions. The Capture-C viewpoint is the *FLT3* TSS. **i** TOPmentation and ATAC-seq at the *FLT3* locus in chALL patient #3. **j** Schematic showing the principle behind the TetR recruitment system. **k** ChIP-qPCR for the indicated proteins (*left*) at the *TetO* array inserted into mESCs expressing the indicated TetR fusion proteins (*top*). Dashed line shows ChIP-qPCR in cells treated with doxycycline for 6 h. Data are represented as mean ± SEM, *n* = 4 independent experiments for TetR FS2 in TetR, TetR FS2 in TetR-CXXC, TetR FS2 in TetR-Paf1, Enl in TetR, Enl in TetR-ENL, Paf1 in TetR, Paf1 in TetR-PAF1; *n* = 3 independent experiments for TetR FS2 in TetR-ENL, TetR FS2 in TetR-DOT1L, Enl in TetR-CXXC, Enl in TetR-Paf1, Paf1 in TetR-CXXC, Paf1 in TetR-ENL, Ssrp1 in TetR, Ssrp1 in TetR-CXXC, Ssrp1 in TetR-ENL, Ssrp1 in TetR-PAF1 and *n* = 2 independent experiments for Enl in TetR-DOT1L, Paf1 in TetR-DOT1L, Ssrp1 in TetR-DOT1L. Source data are provided as a Source Data file. **l** Model indicating direct or indirect in vivo interactions demonstrated in (**k**).

containing region of MLL. Taken together, these data suggest that FACT is a component of both wild-type and MLL-FP complexes, binding directly to the CXXC domain of MLL.

PAF1C has also been shown to interact directly with the CXXC domain of MLL[60,61]. Both FACT and PAF1C are known to travel with RNAPII and enhance transcription[62–65], so one possibility is that these factors can cooperate with MLL-AF4 to maintain enhancer activity. We observed elevated levels of components of PAF1C (PAF1 and LEO1) and FACT (SSRP1) at both intergenic and intragenic MLL-AF4-bound enhancers, suggesting that MLL-AF4 may recruit PAF1C and FACT to enhancers (Fig. 3f–h, Supplementary Fig. 5a). MLL-AF4 appears to be required for PAF1 enrichment, as MLL-AF4 knockdown reduced levels of PAF1 at both intergenic and intragenic MLL-AF4-bound enhancers, arguing for a direct stabilization of the complex at these loci (Fig. 3f, h, Supplementary Figs. 4e and 5a). This is consistent with previous work from our lab, where we observed a reduction of PAF1 binding upon MLL-AF4 or MENIN siRNA treatment in SEM cells[40]. We confirmed the colocalization of MLL-AF4 and PAF1 at enhancers in primary ALL cells, using PAF1 and H3K79me2 TOPmentation (Fig. 3i, Supplementary Fig. 5b, c). These results suggest that the fusion protein may promote enhancer activity by assembling the same transcription-promoting complex as at promoters, both in cell line model systems and primary leukemia cells.

To understand how FACT and other proteins associate with MLL-AF4 enhancers, we turned to a targeted recruitment system to test in vivo interactions (Fig. 3j). We generated mouse embryonic stem cell lines expressing individual components of the MLL-AF4 complex fused to TetR, which binds at an array of *TetO* repeats inserted into the mouse genome[66]. Previously, we have used transient transfection to express low levels of TetR-MLL-AF4, owing to protein toxicity, but we were unable to detect the interaction between MLL and PAF1C, possibly because it is a weaker interaction than that observed with core complex members such as MENIN[40]. To increase the sensitivity of the assay, we created a stable cell line expressing the MLL CXXC domain alone (TetR-CXXC), as well as TetR fusions of other complex components. Using ChIP-qPCR to assess the ability of these proteins to recruit other factors to the *TetO* array, we detected in vivo interactions between TetR-CXXC with PAF1 and SSRP1 (Fig. 3k). Consistent with biochemical analyses[57–61,67–72] we observed multiple reciprocal interactions between ENL, PAF1 and DOT1L (Fig. 3k). Together, this indicates that PAF1C may be localized to MLL-AF4-bound loci by interaction with both the fusion protein itself and other complex components. Strikingly, we saw a similar effect with FACT, where the MLL CXXC domain, ENL, PAF1 and DOT1L all recruited Ssrp1 to the *TetO* array (Fig. 3k, bottom row). FACT has previously been

demonstrated to interact with Paf1 in yeast[73,74]. We verified that the FACT:MLL-CXXC domain interaction occurs specifically through the CXXC-RD1 domain (Supplementary Fig. 5d). Together, these results argue that colocalization of FACT and the elongation machinery with MLL-AF4 may be achieved via multivalent interactions with multiple components of the complex (Fig. 3l).

## PAF1C and FACT are required for enhancer activity, independent of MLL-AF4 binding

In order to assess the role of PAF1C and FACT in enhancer function, we generated SEM degron cell lines where endogenous PAF1 or SSRP1 was tagged with the FKBP12$^{F36V}$ domain[75]. Treatment of cells with the small molecule dTAG-13 resulted in a rapid and dramatic reduction in protein levels (Fig. 4a, Supplementary Fig. 6a). Reference-normalized transient transcriptome sequencing (TT-seq)[76] revealed a significant downregulation of transcription at the vast majority of expressed genes upon dTAG-13 treatment (Supplementary Fig. 6b). Strikingly, degradation of either PAF1 or SSRP1 loss appeared to reduce the frequency of transcription initiation, and PAF1 loss also had an effect on detectable transcripts toward the 3′ end of the gene, consistent with a role in transcription elongation (Supplementary Fig. 6c). However, from these experiments, it is not possible to conclusively determine which stages of transcription were most sensitive to loss of each factor.

To understand the role of PAF1C and FACT at enhancers, we analyzed eRNA transcription. Degradation of either PAF1 or SSRP1 resulted in a dramatic reduction in eRNA levels at MLL-AF4-bound enhancers (Fig. 4b, Supplementary Fig. 6d), suggesting a key role for these factors in enhancer activity. This effect was particularly striking at the intragenic *FLT3/PAN3* enhancer (Fig. 4c). In contrast to the response to MLL-AF4 knockdown (Fig. 2c), eRNA transcription was also reduced at some non-MLL-AF4-bound enhancers (Fig. 4b, Supplementary Fig. 6d), indicating that PAF1C and FACT may have a more general role at enhancers, beyond their association with MLL-AF4. For example, at *LMO4*, we observed a reduction in enhancer activity (eRNA transcription and H3K27ac) at non-MLL-AF4-bound sites following PAF1 or SSRP1 degradation (Supplementary Fig. 6e). Indeed, there was only a slight, although statistically significant, difference in the extent of downregulation of genes associated with an MLL-AF4-bound or -unbound enhancer (Supplementary Fig. 6f). As with eRNA transcription, levels of H3K27ac were reduced at enhancers following degradation of PAF1 or SSRP1, irrespective of MLL-AF4 binding (Fig. 4c, d, Supplementary Fig. 6e, g). These decreases were observed at intergenic as well as intragenic enhancers, arguing that this is not an indirect effect of loss of gene transcription (Supplementary Fig. 6h).

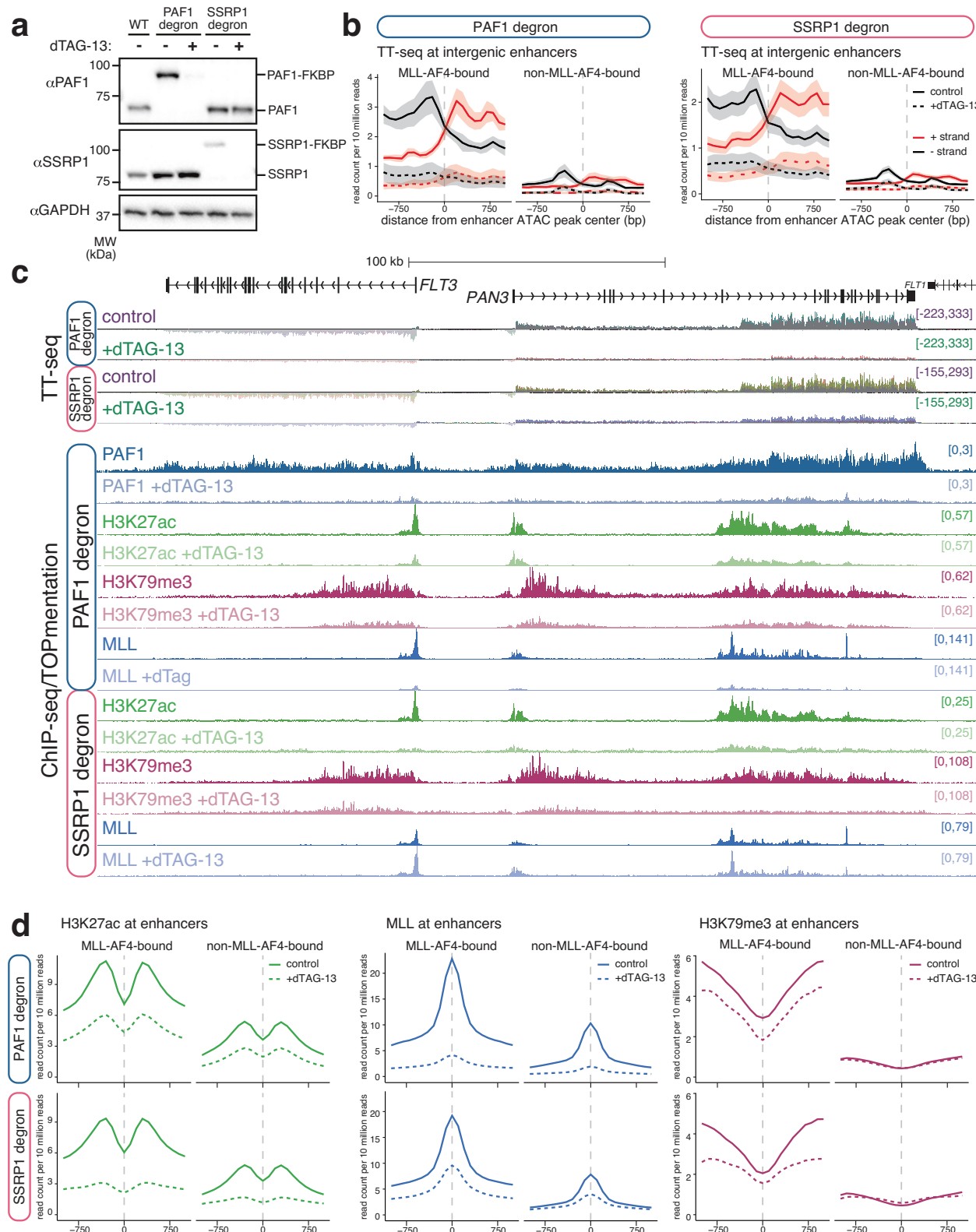

**Fig. 4 | PAF1 and SSRP1 are required for the activity of MLL-AF4-bound and non-MLL-AF4-bound enhancers. a** Western blot for PAF1 or SSRP1 in wild-type (WT), PAF1 degron or SSRP1 degron cells, with (+) or without (−) addition of 0.5 μM dTAG-13 for 24 h. Blots are representative of three replicates. Bands representing wild-type and FKBP12 F36V-tagged proteins are indicated. Source data are provided as a Source Data file. **b** Mean distribution of strand-specific TT-seq (eRNA) levels at MLL-AF4-bound and non-MLL-AF4-bound intergenic enhancers in PAF1 degron or SSRP1 degron cell lines under control (untreated) and 24 h dTAG-13-treated conditions. Lines represent mean, shading represents ± SEM, *n* = 3 independent experiments. **c** TT-seq and reference-normalized ChIP-seq/TOPmentation at the *FLT3* locus in PAF1 degron and SSRP1 degron SEM cells, with or without the addition of dTAG-13 for 24 h. **d** Mean distribution of H3K27ac, MLL and H3K79me3 at MLL-AF4-bound and non-MLL-AF4-bound enhancers in PAF1 degron (*above*) or SSRP1 degron (*below*) cell lines under control (untreated) and 24 h dTAG-13-treated conditions.

Given the enrichment of PAF1C and FACT at MLL-AF4 enhancers and the multivalent interactions we observed between these proteins and members of the MLL-AF4 complex, we asked whether PAF1C and FACT might themselves stabilize the binding of MLL-AF4. Indeed, we observed a decrease in MLL binding at enhancers following the loss of PAF1 or SSRP1, with a particularly striking reduction associated with PAF1 degradation (Fig. 4c, d, Supplementary Fig. 6g, h). H3K79me3 levels were also reduced at MLL-AF4 enhancers, suggesting a loss of DOT1L association/activity (Fig. 4c, d, Supplementary Fig. 6g, h). Thus, we hypothesize that PAF1C and FACT may contribute to enhancer function both directly (by promoting eRNA transcription) and indirectly (via MLL-AF4 complex stabilization).

In order to test the contribution of PAF1 in enhancer activity beyond MLL-AF4 enhancers, we investigated its role in multiple myeloma, which is known to be driven by oncogenic enhancer activity[77–80]. Multiple myeloma is a plasma cell malignancy originating from a more differentiated B lineage cell type than MLL-AF4-driven ALL. We found that PAF1 binds at both typical enhancers and SEs in the multiple myeloma cell line MM1.S (Supplementary Fig. 7a), for example, at the *CCND2* locus (Supplementary Fig. 7b), suggesting it may also have a functional role at enhancers in this disease context. To test this, we generated a pool of MM1.S cells expressing the PAF1 degron, which we used to deplete PAF1 levels (Supplementary Fig. 7c). PAF1 degradation resulted in a reduction in gene transcription (Supplementary Fig. 7d). However, in contrast to SEM cells, where we observed a reduction in both initiation and elongation stages of transcription (Supplementary Fig. 6c), PAF1 degradation in MM1.S cells disrupted transcriptional elongation with no apparent effect on initiation (Supplementary Fig. 7d). We observed only a subtle reduction in eRNA transcription (Supplementary Fig. 7e), and little or no change in enhancer H3K27ac levels (Supplementary Fig. 7b, f, g). Taken together, this suggests that the role of PAF1 in driving enhancer activity is not crucial in all disease contexts and may be unique to MLL-AF4-driven leukemias.

## Enhancer-promoter contacts are differentially dependent on MLL-AF4, PAF1C and FACT

As we had observed a colocalization of MLL-AF4 binding and enhancer-promoter interactions (Fig. 1), we hypothesized that MLL-AF4 and the complex of proteins it assembles at enhancers might play a role in driving these contacts. We used Next Generation Capture-C[81,82] to explore this relationship and test whether proximity was dependent on the presence of MLL-AF4. As with MCC, we observed a striking correlation between MLL-AF4 binding at enhancers and promoter interaction frequency (Fig. 5a, Supplementary Fig. 8a, purple shading). This is particularly clear when comparing the interaction profiles at the same gene in SEM and RS4;11 cells; in each cell type, promoter interaction matches the distinct binding profile of MLL-AF4 (Supplementary Fig. 8b).

MLL-AF4 knockdown significantly reduced the frequency of interactions between MLL-AF4-bound enhancers and promoters of oncogenes such as *ARID1B, CDK6, FLT3, BCL11A* and *PROM1* (Fig. 5a, Supplementary Fig. 8a). Conversely, no change in enhancer-promoter interaction frequency was observed at non-MLL-AF4 enhancer-associated genes such as *FOS, LMO4* and SPI1 (Supplementary Fig. 8c). To broaden this observation, we analyzed a panel of 32 genes, including genes associated with MLL-AF4-bound or non-MLL-AF4-bound enhancers (Supplementary Table 2), and found a clear and specific effect of MLL-AF4 knockdown on promoter interactions with MLL-AF4-bound enhancers (Fig. 5b, Supplementary Fig. 8d). This argues that MLL-AF4 binding at the enhancer is required to drive and/or stabilize contact with the promoter.

We next asked whether the recruitment of elongation factors to enhancers by MLL-AF4 could provide a mechanism by which the fusion protein achieves enhancer-promoter contact. Many enhancer-promoter interactions were sensitive to degradation of PAF1 or

SSRP1 (Fig. 5a, b, Supplementary Fig. 8a), indicating a dependence on PAF1C or FACT for proximity. However, in contrast to MLL-AF4 knockdowns, there was less bias toward MLL-AF4-bound enhancers, with reductions in promoter interactions also observed at non-MLL-AF4-bound enhancers (Fig. 5b, Supplementary Fig. 8c). This is consistent with the observation that eRNA and H3K27ac levels at both MLL-AF4 and non-MLL-AF4-bound enhancers were sensitive to loss of PAF1 and SSRP1 (Fig. 4b, d). However, this is unlikely to be a general consequence of transcriptional disruption, since degradation of BRD4 has minimal impact on (or even subtly increases) enhancer-promoter interactions at the same set of genes (Fig. 5b)[26]. Together, these data suggest that PAF1 and FACT may have a general role in enhancer activity in MLL-AF4 leukemia cells, including enhancer-promoter contact, but they have a particularly strong impact on MLL-AF4-bound enhancers.

MLL-AF4-bound enhancers are marked with H3K79me3 (Fig. 2f), and the majority of these enhancers are annotated as KEEs (Supplementary Fig. 4g). We have previously shown that KEE-promoter interactions are sensitive to chemical disruption of H3K79me2/3 by DOT1L inhibition[28], and these interactions were also perturbed by MLL-AF4 knockdown (Supplementary Fig. 8d). Knockdown of MLL-AF4 reduced H3K79me3 levels at enhancers (Fig. 2f) suggesting that DOT1L enrichment may provide a potential mechanism for stabilizing enhancer-promoter interactions. Comparing the effect of MLL-AF4 knockdown and DOT1L inhibition revealed a positive correlation (R = 0.77) in the change in interaction frequency between MLL-AF4-bound enhancers and promoters (Fig. 5b, c, Supplementary Fig. 8e), indicating that loss of H3K79me2/3 partially reproduces the effect of MLL-AF4 depletion at these loci. The much weaker correlation between MLL-AF4 knockdown and PAF1 or SSRP1 degradation (Fig. 5c, Supplementary Fig. 8e) suggests that these factors have distinct roles at a subset of MLL-AF4-bound and non-MLL-AF4-bound enhancers.

## MLL-AF4 binding establishes an environment encouraging transcription factor binding

The results presented so far argue that MLL-AF4 engages a network of multivalent interactions, assembling a complex of proteins to maintain high levels of activity at a subset of enhancers, including enhancer-promoter proximity. The broad domains of MLL-AF4 binding at enhancers correlate with a high degree of open chromatin (Fig. 1g, Supplementary Fig. 3d), suggesting extensive transcription factor (TF) occupancy. We therefore hypothesized that MLL-AF4 binding at these loci might cooperatively maintain dense TF binding. To investigate this, we performed an analysis of the genome-wide binding of MLL-AF4 and TFs.

We used motif enrichment to identify whether specific TFs were associated with MLL-AF4-enhancers. While several TF motifs were found to be statistically enriched at MLL-AF4-bound enhancers, including motifs for MEF2D, ATF3 and CREB1, the proportion of enhancers containing these motifs was broadly similar globally at all enhancers, suggesting that MLL-AF4-bound enhancers are unlikely to be defined by specific TF binding sequences (Supplementary Fig. 9a). However, we hypothesized that the increased accessibility associated with MLL-AF4 binding might promote TF binding by increasing sequence availability. To explore this possibility, we focused on RUNX1 and MAZ, which we recently identified as key nodes in MLL-AF4 gene regulatory networks[83], with an important role in regulating gene expression downstream of MLL-AF4. Both TFs showed a high density of binding at MLL-AF4-bound enhancers genome-wide (Fig. 6a), correlating with chromatin accessibility, for example, at *ARID1B* (Fig. 6b).

We have previously shown that at a subset of target genes, MLL-AF4 binding at the promoter spreads into the gene body[40] and that these broad binding domains act as intragenic MLL-AF4-bound enhancers, as at *ARID1B* and *CDK6*[28]. Genes with the broadest MLL-AF4 domains showed much denser RUNX1 and MAZ binding

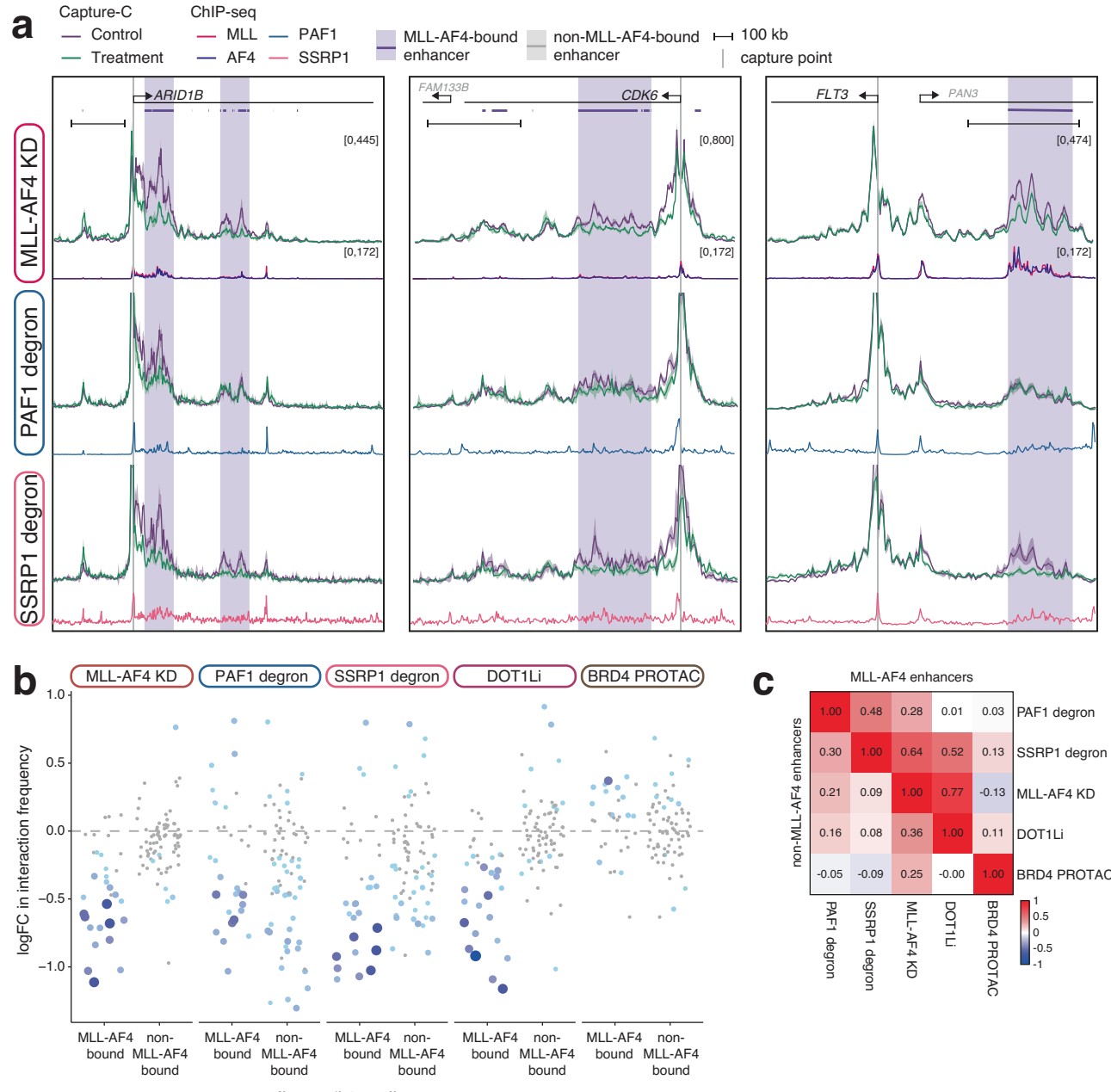

**Fig. 5 | MLL-AF4 binding is necessary to maintain enhancer-promoter interactions. a** Capture-C from the promoters of *ARID1B*, *CDK6*, and *FLT3* in SEM cells under control (purple) and 96 h MLL-AF4 knockdown (green) conditions (*upper*) or in PAF1 degron or SSRP1 degron cell lines under control (purple) and 24 h dTAG-13-treated (green) conditions. Lines represent mean, shading represents ± SEM, *n* = 3 independent experiments. ChIP-seq traces for MLL, AF4, PAF1 and SSRP1 are shown, along with bioinformatically-annotated MLL-AF4-bound (purple bars) and -unbound (gray bars) enhancers. **b** Statistical analysis of the changes in Capture-C

interaction frequency between promoters and MLL-AF4-bound or -unbound enhancers following the indicated treatments. Size and color of the dot are proportional to the significance of the change in interaction with each enhancer. *n* = 3; ns: adjusted *p*-value ≥ 0.05; two-sided Mann–Whitney U test. The 7-day DOT1Li data[28] and 24-h BRD4 PROTAC (AT1) data[26] were previously published. **c** Correlation of changes in Capture-C interaction frequency between promoters and MLL-AF4-bound (upper triangle) or -unbound (lower triangle) enhancers, comparing the indicated treatments.

throughout the gene body (Fig. 6c, right, Supplementary Fig. 9b), suggesting that a higher degree of TF occupancy correlates with MLL-AF4 binding. Notably, genes of a similar length showed a higher frequency of TF binding when there was also a broad MLL-AF4 domain present (Fig. 6d, Supplementary Fig. 9c). Indeed, the density of TF peaks across the gene body increased with greater MLL-AF4 coverage (Fig. 6e, Supplementary Fig. 9d). Importantly, while RUNX1 binding frequency increased with MLL-AF4 spreading, RUNX1 motifs did not (Supplementary Fig. 9e).

Together these analyses suggest that broad MLL-AF4 binding domains are associated with increased TF binding, indicating that MLL-AF4 may help maintain TF binding to contribute to aberrant enhancer activation. We tested this dependency by conducting ChIP-seq for RUNX1 and MAZ following MLL-AF4 knockdown. As RUNX1 and MAZ are both positively regulated by MLL-AF4[83], we chose a 48 h time point to minimize impacts on RUNX1 and MAZ protein levels due to gene expression changes. At this time point, MAZ protein levels remained stable, but RUNX1 protein levels were very slightly reduced

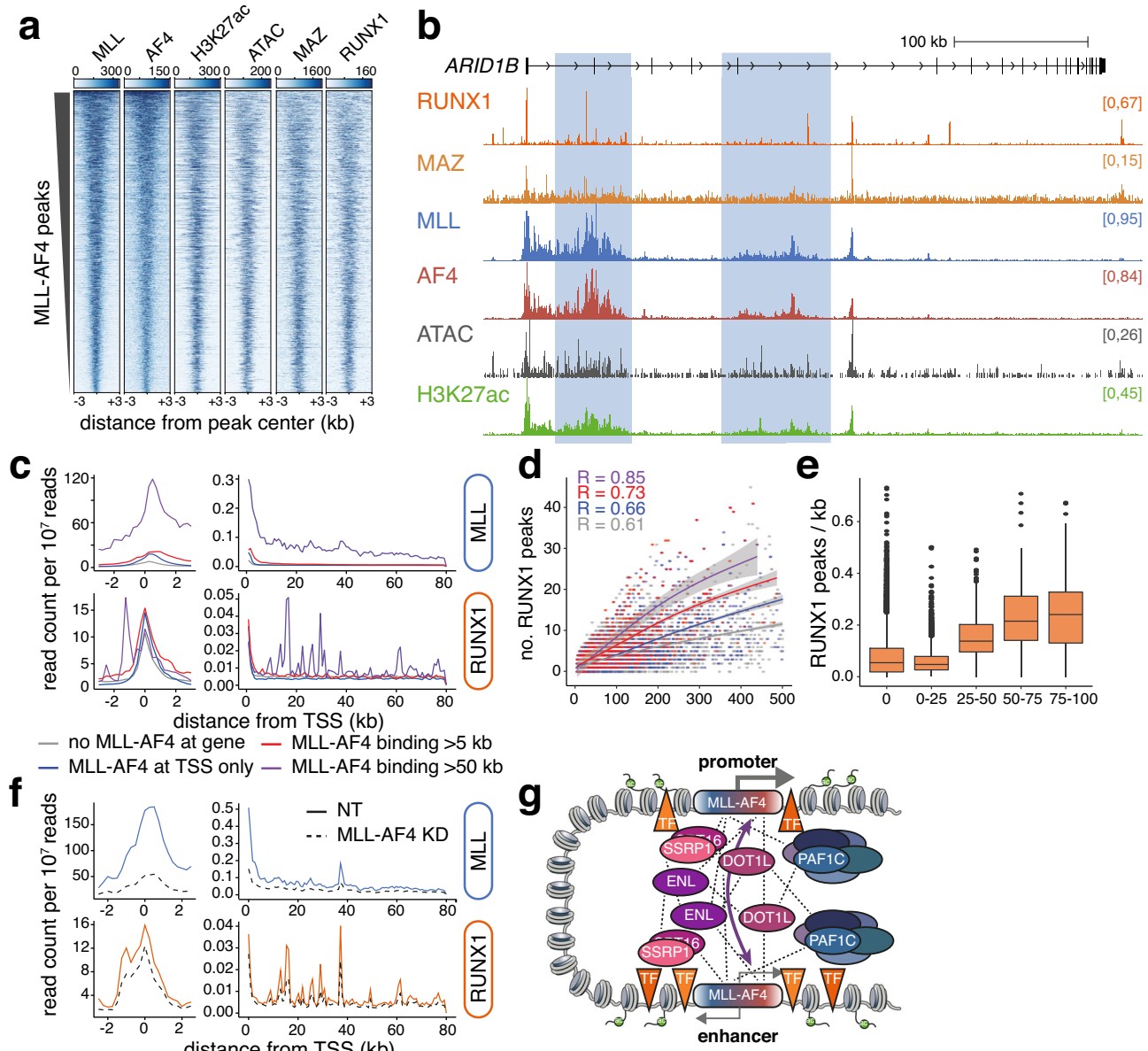

**Fig. 6 | MLL-AF4 binding correlates with high-density TF binding. a** ChIP-seq and ATAC-seq heatmaps at MLL-AF4 peaks in SEM cells. **b** ChIP-seq for RUNX1, MAZ, MLL, AF4 and H3K27ac, and ATAC-seq at *ARID1B*. Putative enhancers are highlighted. **c** Mean ChIP-seq signal at expressed promoters over a 6 kb (*left*) or 80 kb (*right*) window. Profiles stratified by MLL-AF4 binding status. **d** Relationship between RUNX1 peak frequency within gene body and gene body length, stratified by MLL-AF4 binding status as in (**c**). Local regression (LOESS) lines fit are shown, with 95% confidence interval in gray. Correlation (R) calculated by MLL-AF4 binding status. **e** Density of RUNX1 ChIP-seq peaks over gene bodies, stratified by proportion of MLL-AF4 coverage, *n* = 1. Midline shows median, with upper and lower hinges showing the 25th and 75th percentile, respectively. Upper and lower hinges extend to the largest and smallest datapoints within 1.5 times the interquartile range of either hinge. **f** Mean ChIP-seq signal under control (NT) and 48 h MLL-AF4 knockdown conditions, at expressed promoters of genes containing an MLL-AF4 binding domain >50 kb, over a 6 kb (*left*) or 80 kb (*right*) window. **g** Model for the role of MLL-AF4 at enhancers in driving interaction with and transcription of target genes, recruiting a complex of transcription-associated proteins. Dashed lines indicate the network of protein-protein interactions.

(Supplementary Fig. 9f). MLL-AF4 KD resulted in a reduction of RUNX1 binding at broad MLL-AF4 enhancers (Fig. 6f), for example at *ARID1B* (Supplementary Fig. 9g), suggesting that RUNX1 binding at these loci is dependent on MLL-AF4. In contrast, surprisingly, MAZ binding increased at enhancers following MLL-AF4 KD, suggesting potential competitive binding between MAZ and MLL-AF4 (Supplementary Fig. 9h). One possibility is that hypomethylation of MLL-AF4-bound regions[40] partly contributes to increased DNA binding of factors such as MAZ, which generally have a preference for CG rich DNA[84]. In the absence of MLL-AF4, DNA methylation is unlikely to be rapidly re-established, meaning that, without competition, MAZ binding at these loci may increase. Thus, there may be a complex interplay between

MLL-AF4 complex activity and DNA hypomethylation driving the binding of some TFs at MLL-AF4 enhancers.

From this work, we propose the following observations and a model for MLL-AF4-mediated enhancer activity. Our data indicate that PAF1C and FACT have a general function at enhancers in SEM cells, which is not specifically limited to MLL-AF4-bound loci. However, MLL-AF4 generates a particularly high density of enhancer interactions at a specific subset of enhancers associated with *MLL*r-driven oncogenes such as *FLT3*. MLL-AF4 maintains high-density enhancer-promoter interactions at these enhancers by enriching for an array of transcriptional machinery, including elongation factors such as PAF1C and FACT (Fig. 6g). This is driven by a network of multivalent interactions,

creating a high density of chromatin accessibility, associated with TF binding. This clustering of activity may also contribute to the recruitment of additional enhancer factors such as BRD4 and Mediator. Together, these high-density interaction clusters create highly active enhancers that produce increased levels of target gene transcription.

## Discussion

Higher-order 3D genome organization is becoming better understood, but it is still not known how enhancers locate promoters over a distance and come within proximity of the genes they regulate[3,7,8]. Enhancer activation is a common mechanism for gene upregulation in cancer[24,85,86]. We and others have identified key enhancers in *MLL*r leukemias[28,41–43,87], but it has been unclear whether MLL-AF4 has a direct role at these sites and how it may drive aberrant enhancer activity. In this paper, we confirm that MLL-AF4 is required to maintain the activity of many enhancers and identify a role for PAF1 and FACT, factors normally associated with transcription elongation, in helping mediate enhancer-promoter proximity. PAF1 and FACT are highly enriched at enhancers bound by MLL-AF4 in *MLL*r leukemias and help create aberrant regions of high-density promoter interactions. MLL-AF4 enhancers are associated with a high frequency of TF binding, but MLL-AF4 also regulates the expression of a network of TFs, directing a program of leukemic transcription[83], and we note that it is likely that some leukemic enhancers are activated indirectly via the TFs within this network.

Many of the MLL-AF4-bound enhancers are also active in non-*MLL*r ALL cell lines, indicating that these are common regulatory elements in leukemia. There are likely to be distinct, MLL-AF4-independent mechanisms maintaining these enhancers in non-*MLL*r ALL, and it is also possible that MLL-AF4 is not the main driver of enhancer activity at these loci in MLL-AF4 leukemia. In contrast, we also identified a subset of MLL-AF4-bound enhancers that were unique to *MLL*r ALL, which were associated with key oncogenes showing *MLL*r-specific transcriptional upregulation. Many of these MLL-AF4-unique enhancers were also absent from normal hematopoietic cells, indicating that they likely arise de novo during leukemogenesis. A strong example of this is the *FLT3* enhancer, which is not as large or extensive in either normal cells or other ALL samples compared to MLL-AF4 leukemias. This strongly suggests that MLL-AF4 binding may be responsible for the formation of these enhancers; however, we note that from the data presented in this paper, we are only able to conclude a role for MLL-AF4 in enhancer maintenance. Direct targeting experiments will be required to demonstrate whether MLL-AF4 itself can generate enhancers de novo.

Remarkably, loss of PAF1C or FACT resulted in a reduction in the frequency of interaction between specific enhancer and promoter loci, indicating roles for these factors in 3D contacts. In contrast, despite having a prominent role at enhancers, inhibition or degradation of BRD4 significantly downregulates gene expression without affecting enhancer-promoter contacts[26]. Thus, disrupting transcription alone is not enough to impact enhancer-promoter proximity, suggesting additional roles for PAF1C and FACT at enhancers. Similarly, many of the effects of MLL-AF4 knockdown on enhancer activity were replicated by DOT1L inhibition[28], suggesting that MLL-AF4 acts via the localization of multiple factors to enhancers to drive transcriptional upregulation. Degradation of PAF1 or SSRP1 produced distinct but overlapping effects on enhancer activity compared to MLL-AF4 knockdown. This hints at roles at enhancers independent of MLL-AF4 and therefore possibly beyond *MLL*r leukemia. It is possible that context plays an important role in determining which enhancers are dependent on PAF1C and FACT.

While PAF1 was originally identified as a transcriptional elongation factor, it has since been found to regulate multiple stages of the RNAPII transcription[62,88]. Perhaps because of this pleiotropy, in vivo studies of PAF1C function have reported contradictory effects of PAF1 depletion

on gene expression. In some, knockdown of PAF1 increased RNAPII and super elongation complex (SEC) occupancy across gene bodies, suggesting a role in suppressing promoter-proximal pause release[89,90]. However, other studies instead showed that loss of PAF1C impaired transcription elongation, indicating that PAF1 was essential for productive promoter-proximal pause release[91–95], consistent with in vitro transcription and structural studies[63,96,97]. It is possible that PAF1C has different roles at different genes or in different cellular contexts; our results here are consistent with PAF1C promoting transcription elongation in both MLL-AF4 ALL and multiple myeloma cells.

Recently, the use of degron technology has allowed the consequences of acute PAF1C degradation to be investigated. One study in K562 (chronic myelogenous leukemia) cells identified distinct requirements for different PAF1C components in stimulating RNAPII activity, with degradation of the PAF1 subunit only weakly affecting RNA synthesis[95]. This is in contrast to the dramatic reduction in transcription produced by PAF1 degradation in SEM (MLL-AF4 ALL) cells but matches the much weaker effect we observed in MM1.S (multiple myeloma) cells. Thus, while PAF1 may have a particularly central role in transcription in MLL-AF4 leukemias, this is not universal and may be a consequence of the activity and multivalent interactions of the MLL-AF4 complex. A full answer to this question will be revealed by studying PAF1C function in a wider range of cell types and tissues.

While relatively understudied, roles for PAF1C at enhancers have also been proposed[98–100]. Our finding that loss of PAF1C reduced enhancer activity, not only at MLL-AF4 bound enhancers but also at many non-MLL-AF4 bound enhancers, complements a recent study showing a strong correlation between PAF1C binding and enhancer activity[99], arguing for a role in promoting transcription. In contrast, Chen et al.[98] proposed that PAF1C restrains activation of a subset of enhancers by inhibiting RNAPII promoter-proximal pause release. Another study found that PAF1 knockout resulted in increased eRNA levels, which the authors attributed to PAF1 recruitment of Integrator to terminate transcription[100]. In our work, we observe the opposite effect; PAF1 degradation significantly decreased eRNA transcription in SEM cells. One possibility is that the timing of PAF1 depletion is critical; our assays were conducted after 24 h, whereas both previous studies used a window of 72 h or longer[98,100], which could allow for additional secondary effects on enhancer activity. Surprisingly, while we observed a strong effect of PAF1 depletion on enhancer function in SEM cells, the consequences were minimal on MM1.S enhancer function, despite the high frequency of PAF1 binding at enhancers. This may also explain why SEM cells displayed a clear reduction in transcription initiation, whereas MM1.S cells only showed an effect on elongation. Together, these observations support the notion that PAF1C activity may vary based on the cell type, perhaps dependent on the presence or absence of specific complex components or the chromatin context itself.

Although PAF1C binding to enhancers has been previously observed[98–100], it is not known how it localizes at these sites. Here we show that MLL-AF4 localizes PAF1C to enhancers via multiple direct and indirect interactions in a co-dependent manner, consistent with a role in promoting the transcription of target genes. It is unclear how PAF1C is recruited to enhancers independent of MLL-AF4, perhaps via interaction with RNAPII, although there may also be a role for wild-type MLL[60,61].

The FACT complex functions in transcription by displacing histone H2A/H2B dimers from nucleosomes, promoting access for RNAPII transcription[64,101]. FACT has also been proposed to function in sequestering the H2A/H2B dimer for efficient reincorporation into the nucleosome after RNAPII/PAF elongation-mediated disruption[97]. Consistent with the effects we observe following SSRP1 degradation, depletion of FACT in yeast reduces both transcription and RNAPII assembly at promoters[102]. However, the role of FACT at enhancers is unclear, with some studies arguing it may suppress the expression of

target promoters[103], although our data argue for a role in promoting gene activation. As has been proposed for PAF1C[99], FACT may perform a similar function at enhancers as in the gene body, facilitating RNAPII transcription, in this case, for the production of eRNAs. FACT disruption of nucleosome structure at enhancers may also contribute by aiding the binding of TFs, which in turn drive enhancer features. Indeed, we found that MLL-AF4 enhancers show a high density of chromatin accessibility and TF binding.

To understand how MLL-AF4 assembles a transcriptional complex at promoters and enhancers, we used the TetR binding assay to identify in vivo interactions. DOT1L and H3K79me2/3 are known to be associated with MLL-AF4 target genes, but how they are recruited is unclear. ENL/AF9 can interact with DOT1L via their ANC homology domain (AHD)[69,71,104]. However, this domain is also the site of interaction with the AF4 fusion partner, placing MLL-AF4 and DOT1L in mutually exclusive complexes[58,59,69]. Thus, ENL/AF9 alone cannot mediate the MLL-AF4/DOT1L interaction. Our data demonstrate that DOT1L also interacts with PAF1C, consistent with the fact that Paf1 is required for H3K79 methylation in yeast[105,106]. PAF1C also binds to MLL via the MLL CXXC domain[60,61] and ENL via its YEATS domain[67,107], indicating two mechanisms by which PAF1C could associate with the MLL-AF4 complex. The ability of both the MLL and AF4 portions of the fusion to interact with many of these proteins may explain the strong enrichment of factors at MLL-AF4 binding sites. Thus, a network of multivalent interactions may assemble and stabilize the complex of proteins associated with MLL-AF4 at chromatin without requiring continuous binding. We identified FACT as a component of this complex, interacting with MLL, ENL, PAF1C and DOT1L in vivo via similar multivalent contacts. FACT has previously been found to interact with Oct4[108,109] and TIF1γ/TRIM33[89,103,110], suggesting that this may be a common mechanism to promote transcription.

How might MLL-AF4 facilitate the physical proximity of enhancer and promoter? With MLL-AF4 bound at both loci, the numerous multivalent interactions we observed, as well as direct interaction with wild-type AF4 or potentially other MLL-AF4 molecules[57–59], may be sufficient to bridge the interface. We have previously proposed that these interactions drive spreading of MLL-AF4 in cis from the promoter into the gene body[40], and our data here suggest that they may also facilitate trans interactions between MLL-AF4-bound at the enhancer and promoter (Fig. 6g). The idea that multiple weak interactions might drive complex assembly, stability, and ultimately enhancer function is consistent with recent models proposing the assembly of phase condensates as drivers of enhancer activity[111–117]. Degradation of PAF1 or SSRP1 disrupts MLL-AF4 binding, and DOT1L inhibition leads to loss of DOT1L binding to chromatin[118], suggesting that complex disassembly may explain the loss of enhancer-promoter interactions observed when individual components are either inhibited or degraded. We do not exclude a role for cohesin and CTCF, which are found at many of these loci and have been implicated in enhancer function[16,26,119].

Although model systems are essential to fully define the precise mechanisms that underpin the oncogenic behavior of MLL-AF4, it is important to understand how these reflect primary ALL cells in patients. Characterizing the distribution of MLL-AF4 binding and the chromatin profile of MLL-AF4-bound regions has been severely hampered by the limitations of the experimental methodology applicable to the small numbers of cells available in patient samples. Although several key datasets have been published exploring the transcriptional profiles of MLL-AF4 ALL in primary patient samples[34,36], there are few examples of chromatin analysis, often only in a single patient in the absence of coupled transcription data[40,41,83,120]. CUT&Tag has recently been employed as an approach for generating genome-wide binding profiles in lower cell numbers[120]. In order to study the mechanism of MLL-AF4 activity in a disease-relevant context, we developed an optimized ChIPmentation technique, TOPmentation. This allows analysis

of difficult-to-ChIP factors in low cell number samples, enabling the simultaneous description of multiple histone modifications and transcription factors from a single sample alongside paired ATAC-seq and RNA-seq. With this comprehensive profiling, we demonstrated MLL-AF4 binding at enhancers in patients and strong colocalization with PAF1 at these enhancers. We believe that this dataset will be a valuable resource for research into the transcriptional regulation of *MLL*r leukemia.

## Methods

### Ethics
The research in this paper complies with all relevant ethical regulations. More specifically, ethics oversight is governed by the UK Human Tissue Authority (HTA, www.hta.gov.uk) under the Blood Cancer UK Childhood Leukaemia Cell Bank (now VIVO Biobank) ethical approval: NHS HRA: REC: 16/SW/0219. All patients gave informed consent via a specific CellBank consent form, as found at https://cellbank.org.uk/participants. Informed consent was obtained from all participants or those with parental responsibility, and participants did not receive any monetary compensation. The consent is held by CellBank and the treating hospital.

### Patient samples
Infant (<1 year old at diagnosis) and childhood (1–18 years old) ALL samples were obtained from Blood Cancer UK Childhood Leukaemia Cell Bank (now VIVO Biobank). There were three childhood ALL (chALL #1, #2 and #3) primary samples and one infant (iALL #2) primary sample, one female and three males. The infant primograft was from previously published data on sample L826, a male iALL patient (iALL #1)[40,45]. All ALL samples were anonymized at source, assigned a unique study number and linked. Cryopreserved samples were thawed and immediately used for RNA extraction/ATAC analysis or fixed for ChIP-seq/TOPmentation.

### Cell culture and RNA interference
SEM cells (ACC 546)[121] were purchased from DSMZ (www.cell-lines.de) and cultured in Iscove's Modified Dulbecco's Medium (IMDM) with 10% fetal calf serum (FCS, Gibco) and Glutamax (ThermoFisher Scientific). RS4;11 (CRL-1873) and MM1.S (CRL-2974) cells were purchased from ATCC (www.lgcstandards-atcc.org) and cultured in RPMI 1640 with 10% FCS and Glutamax. Cells were validated by the supplier by STR DNA typing. Mouse ES cells with a TetO-array (TOT2N mESC) were kindly provided by Prof. Rob Klose (University of Oxford) and were grown in DMEM supplemented by with 10% FCS, NEAA, Glutamax, LIF and β-mercaptoethanol. All cell lines were confirmed free from mycoplasma contamination.

siRNA knockdown of MLL-AF4 was conducted as previously described[40], using the following sequences: **NT siRNA**: sense AAAAG-CUGACCUUCUCCAAUG; antisense CAUUGGAGAAGGU-CAGCUUUUCU. **SEM MLL-AF4 knockdown siRNA**: sense AAGAAAAGCAGACCUACUCCA; antisense UGGAGUAGGU-CUGCUUUUCUUUU. **RS4;11 MLL-AF4 knockdown siRNA**: sense ACUUUAAGCAGACCUACUCCA; antisense UGGAGUAGGU-CUGCUUAAAGUCC. Briefly, cells were resuspended at a density of $10^8$/ml, and siRNA was added to a concentration of 250 nM. Cells were subjected to a 10 ms 330 V electroporation using a rectangle pulse EPI 2500 electroporator (Fischer, Heidelberg), after which they were diluted to $10^6$/ml. Cells were either harvested at 48 h, or at this point, a second electroporation was performed, and cells were harvested after a further 48 h for a 96 h knockdown.

### Chromatin immunoprecipitation
Chromatin immunoprecipitation was conducted as previously described[28,40]. Briefly, $10^7$ cells were single-fixed (1% formaldehyde for 10 min) for histone modifications or double-fixed (2 mM

disuccinimidyl glutarate for 30 min, then 1% formaldehyde for 30 min) for transcription/chromatin factors, then lysed with 120 μl SDS lysis buffer (10 mM Tris-HCl pH 8.0, 1 mM EDTA, 1% SDS) and sonicated using a Covaris ME220 (Woburn, MA) to generate 200–300 bp fragments. Insoluble material was pelleted, then the supernatant was diluted 10× and pre-cleared for 30 min at 4 °C with rotation using 5 μl protein A- and G-coupled dynabeads (ThermoFisher Scientific). An input sample was taken, then 2 μg antibody was added to the sample before incubation overnight at 4 °C with rotation. The antibodies used are detailed in Supplementary Table 3. Protein A- and G-coupled dynabeads were used to isolate antibody-chromatin complexes, after which the beads were washed three times with RIPA buffer (50 mM HEPES-KOH pH 7.6, 500 mM LiCl, 1 mM EDTA, 1% NP40 and 0.7% sodium deoxycholate) and once with Tris-EDTA. Samples were eluted with SDS lysis buffer, RNase A- and proteinase K-treated and crosslinks were reversed at 65 °C overnight. DNA was purified by PCR purification column (Qiagen) and analyzed by qPCR, relative to input. PCR primer sequences are given in Supplementary Table 4. For ChIP-seq, libraries were generated using the Ultra II library preparation kit (NEB), then sequenced by paired-end sequencing with a 75-cycle high-output Nextseq 500 kit (Illumina).

For reference-normalized ChIP-seq[122], fixed *Drosophila* S2 cells were added to fixed SEM cells prior to sonication at a ratio of 1:4 S2:SEM. After sequencing, reads were adjusted based on the ratio of dm3:hg19 reads in the input and IP samples for each condition.

## ATAC-seq

ATAC-seq was conducted on $5 \times 10^4$ live cells using Nextera Tn5 transposase (Illumina) as previously described[123]. Libraries were sequenced by paired-end sequencing with a 75-cycle high-output Nextseq 500 kit (Illumina).

## TOPmention

HT-ChIPmentation was conducted as previously described[44]. For TOPmentation, protein A-coupled magnetic beads (10 μl) were incubated with 1 μl of the appropriate antibody for 4 h with rotation at 4 °C in 150 μl Binding Buffer (PBS with 0.5% BSA and 1× protease inhibitor cocktail). The antibodies used are detailed in Supplementary Table 3. Cells were single-fixed (1% formaldehyde for 10 min) for histone modifications or double-fixed (2 mM disuccinimidyl glutarate for 30 min, then 1% formaldehyde for 30 min) for transcription/chromatin factors. Fixed samples were lysed in 120 μl HT-CM Lysis Buffer (50 mM Tris-HCl pH 8.0, 0.5% SDS, and 10 mM EDTA, 1× protease inhibitor cocktail) and sonicated using a Covaris ME220 (Woburn, MA) to generate 200–300 bp fragments. The sonicated chromatin was then incubated with Triton-X100 at a final concentration of 1% for 10 min at room temperature to neutralize the SDS in the lysis buffer. The chromatin was pre-cleared for 30 min at 4 °C with rotation using 5 μl protein A-coupled beads to reduce non-specific binding. The antibody-incubated beads were washed in 150 μl Binding Wash Buffer (PBS with 0.5% FCS), and the pre-cleared chromatin was then added to the antibody-coated beads before incubation overnight at 4 °C with rotation.

Immunoprecipitated chromatin was washed three times with RIPA buffer (50 mM HEPES-KOH pH 7.6, 500 mM LiCl, 1 mM EDTA, 1% NP-40, and 0.7% Na deoxycholate). Beads were transferred to a second tube between the first and second wash to remove non-immunoprecipitated fragments that adhered to the tube. The beads were then washed once with Tris-EDTA and once with 10 mM Tris-HCl pH 8.0. The chromatin was then tagmented by resuspending beads in 29 μl Tagmentation Buffer (10 mM Tris-HCl pH 8.0, 5 mM MgCl₂, 10% dimethylformamide) and adding 1 μl of transposase (Illumina). Samples were incubated at 37 °C for 10 min, and the reaction was terminated by adding 150 μl RIPA buffer. Beads were washed with 10 mM Tris-HCl pH 8.0 to remove any detergent and resuspended in 22.5 μl

ddH₂O. In order to amplify the tagmented chromatin, 25 μl NEBNext Ultra II Q5 Master Mix (NEB) and indexed amplification primers (125 nM final concentration) were added, and libraries were prepared using the following thermal profile: 72 °C 5 min, 95 °C 5 min, (98 °C 10 s, 63 °C 30 s, 72 °C 3 min) × 11 cycles, 72 °C 5 min. Library clean-up was performed using Agencourt AMPure XP beads at a 1:1 ratio. Samples were sequenced by paired-end sequencing on a Next-Seq 500 (Illumina).

## ChIP-seq/TOPmentation and ATAC sequencing analysis

Following QC of FASTQ files by FastQC (v0.12.1; http://www.bioinformatics.babraham.ac.uk/projects/fastqc/), reads were trimmed using trim_galore with Cutadapt (v0.6.10; https://www.bioinformatics.babraham.ac.uk/projects/trim_galore/). Trimmed reads were then mapped to the hg19 reference genome using Bowtie 2 (v2.5.1)[124]. PCR duplicates were removed using picard MarkDuplicates (v3.0.0; http://broadinstitute.github.io/picard). Problematic genomic regions present in the ENCODE Blacklist (https://doi.org/10.1038/s41598-019-45839-z) were removed from the aligned files, and further QC of the aligned files was performed using samtools (v1.17)[125]. As many of the factors that were immunoprecipitated have a mix of both sharp and broad modalities, we used either HOMER (v4.11)[126] or the deep learning-based peak caller LanceOtron (v1.0.8)[127] (with a peak score cut-off value of 0.5) to call peaks. Due to the limited material available, replicates were not performed and therefore an IDR-based method was not used. BigWigs were generated using the deepTools (v3.5.1) bamCoverage command[128], with the flags –extendReads –normalizeUsing RPKM, and visualized in the UCSC genome browser[129].

Putative enhancers were identified by intersecting non-promoter ATAC-seq and H3K27ac peaks. For the patient samples, a common MLL-AF4-bound enhancer set was established from enhancers that overlap with MLL peaks in at least three patients. Metaplots were generated using the Homer function annotatePeaks.pl[126], centered on enhancer ATAC peaks. Super-enhancers were identified using the Homer function findPeaks with the options –style super –minDist 12500 –L 1 and tag directories for H3K27ac, H3K4me1, BRD4 and MED1. Transcription factor motif enrichment in MLL-AF4-bound enhancers was calculated using the Homer function findMotifs.pl, with all enhancers used as background. Transcription factor motifs were identified genome-wide using Homer scanMotifGenomeWide.pl with motifs from the Homer motif database.

For the comparative analysis of enhancer activity, read counts over the common MLL-AF4-bound enhancer set were extracted from H3K27ac TOPmentation or ATAC-seq BAM files for all samples using featureCounts v2.0.2[130]. Read counts were normalized using the variance stabilizing transformation in the pyDEseq2 (v0.3.1) package[131]. Normalized counts were then Z-scored and clustered using the K-Means algorithm. Heatmaps were generated using ComplexHeatmap v3.17.

## RNA-seq, qRT-PCR and TT-seq

RNA was extracted from $1 \times 10^6$ cells with the RNeasy Mini Kit (Qiagen). For qPCR, reverse transcription was conducted using Superscript III (ThermoFisher Scientific) with random hexamer primers (ThermoFisher Scientific), and cDNA was analyzed by Taqman qPCR, using the housekeeping gene *YWHAZ* for gene expression normalization.

For patient RNA-seq, mRNA was isolated from bulk RNA using the NEBNext Poly(A) mRNA magnetic isolation module (mRNA) and used to generate a strand-specific library using the NEBNext Ultra II Directional RNA Library Prep Kit for Illumina (NEB).

TT-seq was conducted as previously described[76]. Briefly, thiouridine-labeled spike-in RNA was generated by in vitro transcription of exogenous plasmid sequences in the presence of 4S-UTP (Jena Bioscience) using the MEGAscript kit (ThermoFisher Scientific). Then, $5 \times 10^7$ SEM or MM1.S cells were treated with 500 μM 4-thiouridine for

5 min before RNA was isolated by Trizol extraction (ThermoFisher Scientific) with the addition of 60 ng spike-in RNA, then purified and DNase I-treated. Labeled nascent RNA was fragmented briefly by sonication (Covaris), then biotinylated with EZ-link biotin-HPDP (ThermoFisher Scientific) and purified by Streptavidin bead pull-down (Miltenyi). Strand-specific libraries were prepared using the NEBNext Ultra II Directional RNA Library Prep Kit for Illumina (NEB). Libraries were sequenced by paired-end sequencing with a 75 (patient samples) or 150 (TT-seq) cycle high-output Nextseq 500 kit (Illumina).

## RNA-seq analysis
Reads were subjected to quality checking by fastQC (v0.12.1) and trimming using trim_galore (v0.6.10) to remove contaminating sequencing adapters, poor quality reads and reads shorter than 21 bp. Reads were then aligned to hg19 using STAR (v2.4.2a)[132] in paired-end mode using default parameters. Gene expression levels were quantified as read counts using the featureCounts function from the Subread package (v2.0.2)[133] with default parameters. The read counts were used to identify differential gene expression between conditions using the DESeq2 (v3.12)[134] package. For comparative expression analysis of *MLL*r vs *MLL*wt samples, featureCounts generated expression counts were normalized using the pyDEseq2 variance stabilizing transformation[131] and normalized counts were compared between the enhancer-associated groups. For TT-seq, spike-in RNA levels were quantified by mapping to a custom genome using featureCounts and used to normalize the output of DESeq2. For metagene analysis, each transcript was scaled by binning the normalized coverage into 600 bins. Upstream and downstream flanking regions (2 kb) were split into 200 equal-length bins. Transcripts were split into quartiles based on length, and median coverage within each bin for each quartile was determined. The median of all replicates for all conditions was determined and log2 transformed after adding a pseudocount of 1.

## Patient datasets and gene expression microarray data
Microarray gene expression data from four large cohorts of patients with ALL were analyzed. These cohorts included the Eastern Cooperative Oncology Group (ECOG) Clinical Trial E2993 (GSE34861) cohort: 191 total samples comprising 78 BCR-ABL1 patients, 6 E2A-PBX1 patients, 25 *MLL*r patients (t(4;11): 17, other *MLL*r: 8), and 82 other B-ALL patients[48]; the Children's Oncology Group (COG) Clinical Trial P9906 (GSE28460) cohort: 207 total samples, 23 E2A-PBX1 patients, 21 *MLL*r patients, 3 RUNX1-ETV6 patients, 155 other B-ALL patients (trisomy 4 or 10 patients)[49]; the St. Jude Research Hospital pediatric ALL clinical trial cohort 2003 (http://www.stjuderesearch.org/site/data/ALL3/): 118 total samples, 15 BCR-ABL1 patients, 18 E2A-PBX1 patients, 20 MLLr patients, 20 RUNX1-ETV6 patients, 17 hyperdiploid patients, 28 other B-ALL patients, 14 T-ALL patients[50]; and the St. Jude Research Hospital pediatric ALL clinical trial cohort 2013 (GSE26281): 127 total samples, 18 BCR-ABL1 patients, 8 E2A-PBX1 patients, 15 *MLL*r patients, 24 RUNX1-ETV6 patients, 26 hyperdiploid patients, 11 CRLF2 patients, 11 ERG patients, 14 other B-ALL patients, and 27 T-ALL patients[51]. Microarray data was normalized with RMA method using Expression Console™ software (v1.1, Affymetrix, Santa Clara, CA) for the Affymetrix arrays HG-U133 plus2 (COG data, n = 207) or NimbleScan software (v2.5, Roche NimbleGen, Madison, WI) for the NimbleGen arrays HG18 60mer expression 385K platform (ECOG data, n = 191). Downstream microarray analysis was performed using R version 2.14.0. Heatmaps were generated with Cluster/TreeView3.0 (http://bonsai.hgc.jp/~mdehoon/software/cluster/software.htm).

## Next Generation Capture-C
Capture-C was conducted as described previously[81,82], using 2 × 10⁷ cells per replicate. Briefly, DpnII-generated 3C libraries were sonicated to a fragment size of 200 bp, and Illumina paired-end sequencing adapters (New England BioLabs, E6040, E7335 and E7500) were added

using Herculase II (Agilent) for the final PCR. Indexing was performed in duplicate to maintain library complexity, with libraries pooled after indexing. Enrichment was performed using previously designed Capture-C probes (Supplementary Table 2)[28], with two successive rounds of hybridization, streptavidin bead pulldown (Invitrogen, M270), bead washes and PCR amplification using the HyperCapture Target Enrichment Kit (Roche). Samples were sequenced by paired-end sequencing with a 300-cycle high-output Nextseq 500 kit (Illumina). Data analysis was performed using CapCruncher v0.2.0[82] (https://doi.org/10.5281/zenodo.6326102), and statistical analysis was performed as described[26,28].

## Micro-Capture-C
Micro-Capture-C was performed as described[8,135]. Briefly, 10⁷ SEM cells were fixed with 2% formaldehyde for 10 min, then quenched with glycine and washed with PBS. Cells were permeabilized with 0.005% digitonin (Sigma Aldrich) for 15 min, then snap-frozen. Thawed cells were pelleted and resuspended in reduced-calcium MNase buffer (10 mM Tris-HCl pH 7.5, 1 mM CaCl₂), then split into three aliquots. Cells were treated with a titration of different micrococcal nuclease (MNase) concentrations (typically 10-30 Kunitz U in an 800 µl volume) for 1 h at 37 °C in a Thermomixer (Eppendorf) at 550 rpm. The reaction was stopped with the addition of ethylene glycol-bis(2-aminoethylether)-N,N,N′,N′-tetraacetic acid (EGTA) to 5 mM and 200 µl was removed to assess the extent of digestion. The remaining cells were pelleted, washed with PBS/EGTA, then processed for end repair, phosphorylation and ligation. Cell pellets were resuspended in DNA ligase buffer, supplemented with 400 µM dNTPs and 5 mM EDTA, before addition of DNA Polymerase I large (Klenow) fragment (NEB) to 100 U/µl, T4 polynucleotide kinase (NEB) to 200 U/µl and T4 DNA ligase (Thermo Scientific) to 300 U/µl. The reaction was incubated for 2 h at 37 °C, then for 8 h at 20 °C, at 550 rpm, then cooled to 4 °C overnight. Both the digested and ligated chromatin were decrosslinked at 65 °C in the presence of proteinase K, then DNA was purified using the Qiagen DNeasy Blood and Tissue Kit. Digestion and ligation efficiencies were assessed by TapeStation (Agilent D1000). Library preparation, indexing and capture were performed as described for NG Capture-C. The probes used for capture are given in Supplementary Table 2. MCC analysis was performed using the MCC pipeline[8] (https://github.com/jojdavies/Micro-Capture-C).

## Immunoprecipitation for mass spectrometry analysis
Nuclear extracts from HEK-293 cells expressing a FLAG- and HA-tagged fragment of the MLL protein (amino acids 1101 to 1978; containing the CXXC domain as well as the PHD fingers, referred to as FLAG-C-PHD) were incubated with anti-FLAG M2 beads (Sigma Aldrich M8823) in 50 mM Tris-HCl pH 7.5, 300 mM KCl, and 20% glycerol. Beads were washed in 50 mM Tris-HCl pH 7.5, 200 mM KCl, 0.05% NP-40, and 20% glycerol, and then three times with the same again but with either 150 mM KCl (lanes 1 and 2) or 300 mM KCl (lanes 3 and 4). Bound proteins were eluted with FLAG peptide. Nuclear extracts from normal HEK-293 cells with no expression construct were used as a control. The eluted material was run on a 4–12% NuPAGE gel and stained with colloidal blue. Bands were separately cut out of the FLAG-C-PHD and control lanes and submitted for mass spectrometry.

## Mass spectrometry analysis
The protein samples were processed and analyzed at the Mass Spectrometry Facility of the Department of Pathology at the University of Michigan. Each lane was cut into 12 equal slices, and in-gel digestion with trypsin followed by LC-MS/MS analysis was performed as described elsewhere[60]. Spectra were searched using the Comet (version 2022.01 rev. 0)[136]. Searches were done with the Uniprot Human reference proteome (UP000005640) with reversed sequences appended as decoys (downloaded May 9, 2022)[137]. Each search was done using

the following parameters: parent mass error ±3 Da; fragment bin ion tolerance ±1.0005; tryptic digestion with up to 1 missed cleavage; variable modification of +16 Da on methionine and a fixed modification of +57 Da on cysteine. Search results were then processed using the Trans-proteomic Pipeline (TPP) tools PeptideProphet and ProteinProphet[138–140]. These tools collapse proteins with shared peptide evidence into protein groups.

## Additional immunoprecipitations and GST pulldowns

Nuclear extracts from HEK-293 cells expressing either FLAG-tagged CXXC or PHD constructs were incubated with anti-FLAG M2 beads (Sigma Aldrich M8823) in 50 mM Tris-HCl pH 7.5, 300 mM KCl, and 20% glycerol. Beads were washed in 50 mM Tris-HCl pH 7.5, 150 mM KCl, 0.05% NP-40, and 20% glycerol. Bound proteins were eluted with FLAG peptide. For GST pull-down assays, 500 ng of GST-fused proteins and 200 ng of purified factors were mixed with glutathione-sepharose 4B beads in binding buffer (20 mM Tris-HCl pH 7.9, 300 mM KCl, no EDTA, 20% glycerol, 0.1% NP-40) and washed in the same buffer. Bound proteins were analyzed by western blotting (for antibodies, see Supplementary Table 3) and/or silver staining.

## Western blotting

For western blotting, proteins were extracted from $1 \times 10^6$ cells using a high-salt lysis buffer (20 mM Tris-HCl pH 8.0, 300 mM KCl, 5 mM EDTA, 20% glycerol, 0.5% IGEPAL CA-630, protease inhibitor cocktail). The antibodies used are detailed in Supplementary Table 3.

## TetR cell lines

For the TetR recruitment assay, we used the previously engineered *Tet Operator* (*TetO*) mESC line[66]. The MLL CXXC domain, ENL, PAF1 and DOT1L cDNA sequences were cloned into the original pCAGFS2TetR vector downstream of the FS2-TetR open reading frame. Plasmids were transfected into mESC using Lipofectamine-2000, and puromycin (1 μg/ml) was used to select for clones stably expressing the TetR construct (confirmed by western blotting). As a negative control, cells were treated with 1 μg/ml doxycycline for 6 h to disrupt TetR binding at the *TetO* prior to fixation for ChIP.

## Generation of degron cell lines

SEM or MM1.S cell lines were edited to fuse both alleles of the *PAF1* or *SSRP1* gene to *FKBP12^{F36V}-P2A-mNeonGreen*, immediately prior to the stop codon. This was achieved by CRISPR/Cas9-mediated homology-directed repair (HDR), as previously described[119]. Cells were electroporated with two plasmids: pX458, encoding Cas9, an sgRNA sequence targeting the end of *PAF1* or *SSRP1* (see Supplementary Table 4) and mRuby; and an HDR plasmid containing the *FKBP12^{F36V}-P2A-mNeonGreen* sequence flanked by 500 bp sequence with homology to either side of the insertion site. After 24 h, mRuby-positive cells were isolated by fluorescence-activated cell sorting to identify cells expressing Cas9 and allowed to grow for 1–2 weeks. mNeonGreen-positive cells were subsequently sorted to isolate cells that had correctly incorporated the insertion sequence within the open reading frame of *PAF1/SSRP1*. After a further 1–2 weeks' growth, cells with the highest mNeonGreen signal were isolated to enrich for homozygous insertions and, to isolate clonal populations, SEM cells were plated onto H4100 Methylcult (StemCell Technologies). Cell lines are available from the corresponding authors upon completion of a Materials Transfer Agreement.

## Reporting summary

Further information on research design is available in the Nature Portfolio Reporting Summary linked to this article.

## Data availability

The raw high-throughput sequencing data generated in this study have been deposited in the Gene Expression Omnibus (GEO) under accession number GSE202451 (SEM/ALL; polyA-RNA-seq, ChIP-seq, ATAC-seq, TOPmentation, TT-seq, Capture-C and MCC data) and GSE236664 (Multiple Myeloma; ChIP-seq and TT-seq data). The raw mass spectrometry data generated in this study were deposited to the ProteomeXchange Consortium via the PRIDE partner repository with the dataset identifier PXD043920. The publicly available cell line ChIP-seq data reused in this study are available in GEO under accession codes GSE74812[141], GSE83671[40], GSE71616[142], GSE151390[83], GSE95917[143], GSE44931[24], GSE95511[144], GSE79899[43], GSE117865[28], GSE84116[145] and GSE186941[46]. The publicly available ATAC-seq data reused in this study are available in GEO under accession codes GSE117865[28] and GSE74912[53]. The publicly available SEM nascent RNA-seq data reused in this study are available in GEO under accession code GSE85988[42]. The publicly available Next Generation Capture-C data reused in this study are available in GEO under accession codes GSE117865[28] and GSE139437[26]. The publicly available ALL patient ChIP-seq data reused in this study are available in GEO under accession codes GSE135024[41], GSE151390[83] and GSE83671[40]. The publicly available ALL patient RNA-seq data reused in this study are available from the European Nucleotide Archive under accession PRJEB23605[36] and microarray expression data are available in GEO under accession codes GSE34861[48], GSE11877[49], GSE26281[51] and at http://www.stjuderesearch.org/data/ALL3[50]. The remaining data are available within the article, Supplementary Information or Source Data file. Source data are provided with this paper.

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

## Acknowledgements

T.A.M., N.T.C., A.L.S., L.G., A.M.D.-F., S.Rice and N.E.J. were funded by Medical Research Council (MRC, UK) Molecular Haematology Unit grant MC_UU_00016/6 and MC_UU_00029/6. N.T.C. was supported by a Kay Kendall Leukaemia Fund Intermediate Fellowship (KKL1443). C.C. was supported by a Wellcome Trust Genome Medicine and Statistics studentship. J.R.H. was funded by an Engineering & Physical Sciences Research Council (EPSRC) Doctoral Training Program grant project, reference 2119788 and EP/R513295/1. A.R. was supported by a Bloodwise Clinician Scientist Fellowship (grants: 14041 and 17001) and Wellcome Trust Clinical Research Career Development Fellowship (216632/Z/19/Z). I.R. was supported by the NIHR Oxford BRC, by a Bloodwise Program Grant (13001) and by the MRC Molecular Haematology Unit (MC_UU_12009/14). R.G.R. was funded by a Leukemia and Lymphoma Society Specialized Center of Research Grant (7021-20). Samples and data used in this study were provided by VIVO Biobank, supported by Cancer Research UK & Blood Cancer UK (Grant no. CRCPSC-Dec21\100003). This research was funded in whole, or in part, by the UKRI [MC_UU_00016/6, MC_UU_00029/6, MC_UU_12009/14 and EP/R513295/1]. For the purpose of Open Access, the author has applied a CC BY public copyright licence to any Author Accepted Manuscript version arising from this submission.

## Author contributions

N.T.C., A.L.S. and T.A.M. conceived the experimental design. N.T.C., A.L.S., L.G., A.M.D.-F., N.D., N.E.J., S.Rice, J.K., V.B. and T.A.M. carried out experiments. N.T.C., A.L.S., J.R.H., J.C.H., C.C., S.Riva, D.F., V.B. and H.G. analyzed and curated the data. N.T.C., A.L.S. and T.A.M. interpreted the data and wrote the manuscript. R.G.R., C.D.A., A.R., I.R., J.O.J.D. and T.A.M., provided funding and supervision. All authors reviewed the manuscript.

## Competing interests

T.A.M. and N.T.C. are paid consultants for and shareholders in Dark Blue Therapeutics Ltd. J.O.J.D. is a founder of and consultant for Nucleome Therapeutics. The remaining authors declare no competing interests.
