## [Peer Review File · Nature Communications]

MLL-AF4 cooperates with PAF1 and FACT to drive high density enhancer interactions in leukemiaReviewers' Comments:

Reviewer #1:

Remarks to the Author:

Crump and Milne et. al. have performed a detailed study evaluating the role of MLL::AF4 (KMT2A::AFF1) within the regulation of enhancers in acute lymphoblastic leukemia (ALL). The authors combined analysis of five primary MLL::AF4 positive pediatric ALL cases with more detailed analyses of the MLL::AF4 positive cell lines SEM and RS4;11. Their data suggest that enhancer-binding of MLL::AF4 is involved in generating interactions between enhancers and promoters. In addition, the elongation factors PAF1C and FACT are suggested to have an important role within the enhancer-promoter interactions. The authors thus presents further details of the function of MLL::AFF1 within leukemia.

The data are clearly presented, methodology and the presentation thereof is sound, and this report is of potential interest. My concerns with this study are outlined below.

1. Currently, the only non-MLL-rearranged-associated data are presented in Extended Data Figure 1h, with database-derived ATAC-seq data on various normal blood/bone marrow precursor cells for the FLT3 enhancer. Hence, no non-MLLr primary ALL-patient samples and/or ALL cell lines were investigated as a control to check for MLLr-specificity versus if some of the findings (e.g. open chromatin, H3K27ac status, etc.) are commonly found in ALL at these locations, potentially aided via wild type MLL at the same enhancer(s) in a leukemia setting. If the authors cannot access any primary non-MLLr samples, it would be good to at least see corresponding results with regards to the status of various enhancers for a couple of ALL cell lines of other subtypes than MLLr.

2. What is the overall correlation between enhancer-occupancy by MLL::AF4 and transcriptional output from the gene predicted to be regulated by the respective enhancer? Consider using publicly available transcriptomics data on pediatric ALL for analysis of differential expression between MLL::AF4 ALL vs. non- MLL::AF4 ALL (including/excluding cases with other MLL fusion partners), with a specific focus on the overlap of MLL::AF4-associated enhancers found in the five primary ALL cases. Also, how large is that overlap?

3. It is stated that gene expression data exist for 4/5 primary ALL specimens, but these data are not used within the current study. These data could, however, be used to investigate transcriptional output of genes here suggested to be regulated via the action of MLL::AF4 on the associated enhancers. Are all the respective genes (highly) expressed in these cells, i.e. is the predicted effect on transcriptional output actually seen? Linking back to both of the comments above: Albeit with a rather small sample number, a comparison between MLLr and non-MLLr cases with regards to transcriptional output of the specific genes could also be performed if the non-MLLr samples mentioned in comment #1 also were to be investigated via RNA-seq.

4. Western blot and/or qPCR data should be presented with regards to siRNA-mediated knockdown efficiency.

Minor:

1. Should it be "Fig. 6g" on page 15, in the sentence associated with facilitation of "trans interactions between MLL-AF4 bound at the enhancer and promoter"?

2. Please provide a full reference for the HT-ChIPmentation in the TOPmentation section in the Methods.

Reviewer #2:

Remarks to the Author:

In their manuscript "MLL-AF4 cooperates with PAF1 and FACT to drive high density enhancer interactions in leukemia" Crump et al. comprehensively analyze the enhancer landscape and the enhancer protein binding patterns in MLL-AF4 fusion driven leukemias. The authors demonstrate that MLL-AF4 binding creates enhancer-promoter interactions and that the transcription elongation complexes PAF1C and FACT interact with MLL-AF4 to promote this function.

The work presented here is a very thorough and technically skilled work combining latest omics technologies with functional assays. The authors also optimized the chipmentation protocol so that they were able to analyze TF binding in primary samples with as few as 100k cells as input. Although, practically all of the functional data were (for good reasons) generated in cell lines, the authors provide convincing data that the proposed interaction partners of MLL-AF4 actually colocalize in primary human leukemia samples.

In summary, I think that this manuscript is well written and the work it describes is important, impactful and technically well-done. I only have a few suggestions that might help to further substantiate some of the claims made in the manuscript.

- Page 5: "We observed a similar distribution in a range of MLLr ALL and AML cell lines (Extended Data Fig. 1d), indicating that non-promoter binding, possibly at enhancers, is a widespread property of MLL-FPs,..." I would suggest to formally test whether MLL-AF4 peaks show enrichment in enhancer regions.
- Extended Data Figure 1c: It is not entirely clear how the correlation was done, i.e. does this mean number of reads in peaks? Also how was this normalized? What is the actual concordance in peaks calls? It might be a good idea to follow the standard QC procedure as suggested by the ENCODE project: <https://www.encodeproject.org/chip-seq/transcription-factor-encode4/>
- Page 5: "We saw a strong correlation between levels of H3K27ac and MLL-AF4 binding at both promoters and enhancers (Fig. 1c, e), indicating that MLL-AF4 associates with enhancer as well as promoter activity in primary ALL cells.": Why is the correlation between MLL binding and H3K27ac weaker at enhancers than at promoters? Is this expected?
- Page 6: "This increased accessibility appears to be associated with leukemogenesis, as, for example, the high density of ATAC-seq peaks observed at the FLT3 enhancer in MLL-AF4 patients and cell lines is completely absent in normal cells, including those from the B lineage (Extended Data Fig. 1h)": I would suggest to support this statement by a more systematic (genome-wide) analysis, e.g. by showing the read coverage at MLL-enhancers. Similarly, I suggest to support the statement related to the MCC interactions which relate to Fig 1g with a more systematic, genome-wide analysis.
- Page 6: "We intersected SEM enhancers and MLL-AF4 peaks, identifying enhancers enriched (bound) or depleted (not bound) for MLL-AF4 (Fig. 2a), and linked these to the nearest gene.": Why were the enhancers linked to the closest gene? The authors have data for gene promoters interacting with the individual enhancers. Why didn't they use this information? I suggest that this should be done in a systematic, data-driven way.
- Page 12: "Together this suggests that broad MLL-AF4 binding domains are associated with increased TF binding, indicating that MLL-AF4 may promote TF binding to contribute to aberrant enhancer activation.": To confirm this finding, I suggest that the authors perform RUNX1 and MAZ1 ChIP-seq after MLL-AF4 KD. If MLL-AF4 is the driver for TF binding events as the presented data suggest, then TF peaks should disappear upon MLL-AF4 KD.

Reviewer #3:

Remarks to the Author:

This study by Crump et al seeks to identify mechanisms of MLL-AF4 mediated enhancer activity in leukemia cells. The authors used patient samples as well as the MLL-AF4 cell line, SEM, to profile chromatin binding, enhancer-promoter interactions, and gene expression in the context of MLL-AF4 knock-down and PAF/FACT complex depletion. This group has published many of the key papers describing MLL-AF4 binding in leukemia cell lines and the current study represents a natural extension of their prior work. The authors used extensive perturbation approaches to further refine principles of MLL-AF4 chromatin binding and generated a plethora of genomics datasets that will be useful to the MLLr/leukemia research community. Of note, the current study developed an improvement to the ChIPmentation technique (termed TOPmentation) to profile chromatin occupancy in 100k primary cells, generating MLL-AF4/histone mod/PAF1/FACT binding in 5 patient samples that identified putative MLL-AF4-generated de novo enhancers. This study also demonstrated a convincing role for MLL-AF4 in mediating enhancer-promoter chromatin interactions and provided early evidence that collaboration with transcription elongation machinery at enhancers may be important for MLL-AF4 mediated enhancer regulation. Technically, this study was sound, especially with the use of acute protein degradation to study the impact of PAF1/FACT loss on enhancer structure. Conceptually, many of the findings reported are not completely novel (MLL-AF4 binds enhancers; the 'spreading' pattern of MLL-AF4 binding; co-occupancy with PAF1, ENL, MENIN; MLL-AF4 having a positive effect on gene expression; correlation with H3K27ac (PMID 28076791). However, through the use of perturbation studies including MLL-AF4 knockdown and acute PAF1/SSRP1(FACT) degradation, the authors demonstrate that MLL-AF4 is required to maintain these features of enhancer activity which had been suggested by prior correlative studies and implicates transcription elongation components in this function. The finding that MLL-AF4 is required for enhancer promoter contact is significant. Future work will undoubtedly resolve whether PAF1/FACT and MLL-FP association has a leukemia-specific role or is reflective of an emerging role for these components at enhancers, in general. The extensive chromatin profiling datasets generated will be a valuable resource to the broader leukemia research community.

Major points

- Key experimental details are missing and should be added to the current manuscript even if reported in the authors' prior publications, including data on how well the MLL-AF4 knockdown worked (western blot), and the time post-perturbation (siRNA transfection or dTAG13 treatment) of conducting RNA-seq/ChIP-seq/Capture-C. The only indication is a reference to their prior study for MLL-AF4 knock-down, where they reported waiting 24 hours before conducting an assay post knock-down. Is that the same here? It seems like it may be too short for protein loss due to RNA knockdown to impact global chromatin structure changes reported here, especially as the authors acknowledge that differences in treatment time may underlie conflicting results observed in this study and a prior study (ref80 10.1016/j.molcel.2021.05.028) Clarifying the critical temporal component of each experiment is highly recommended.
- There are several points of contradiction with both the authors' and others' prior work, most notably regarding the role of PAF1. This is not adequately addressed in the discussion, especially the study by Chen et al (PMID 28860207) which showed that PAF1 loss is an important step in enhancer activation using human cancer cell lines, in contrast to this current study where the authors show that PAF1 loss leads to enhancer decommissioning. Prior literature also showed that PAF1 regulates PolII pause release, and PAF1 depletion led to increased transcription due to release from paused Pol II therefore the results in the current study are confusing as they see that PAF1 depletion leads to a reduction in transcription. The authors discuss these contradictions only superficially and conclude that the differences may simply be due to a difference in treatment time (again highlighting the need to clarify each treatment design in this current study).

Minor points

- MLL-AF4 binding in patient samples is an important technical achievement and the authors state that enhancer binding may help determine the biology of the leukemias. The patient data seem under-utilized, especially the observation of putative MLL-AF4-driven de novo enhancers (e.g. in PAN3). How

many promoters are bound by MLL-AF4 in primary leukemia cells and are they all active? What is the correspondence between primary cell binding and SEM binding? How many putative de novo enhancers were found? What is the correspondence between the 5 patients? Which TF motifs are enriched in the ATAC-seq peaks of leukemia-specific MLL-AF4 bound enhancers? What are their presumed target genes? These are merely suggested analyses to bolster the use of the patient data.

- The plots used to show enrichment and/or change in read counts for different factors are good summaries (“mean distribution”) but may oversimplify the data (it’s not possible to gauge the distribution of fold changes observed at all peaks by reducing the data to a single line showing the mean). At least for a few key figures (2c, 4d), having a volcano plot or violin/box plot that shows the fold change of reads for each peak analyzed (i.e. all show all data points) may help better show relationship and could point to sets of enhancers that are more sensitive to MLL-AF4 loss (in terms of PAF1 binding or histone mod) than others. For example, in Figure 2c it is claimed that MLL-AF4 bound enhancers show a greater loss of H3K27ac compared to non-MLL-AF4 bound enhancers following MLL-AF4 knockdown, however the non-MLL-AF4 bound enhancers have a lower H3K27ac signal to begin with; thus a more fair visual display would indicate differences in fold-changes rather than total read counts. This would also clarify whether there is truly a “much larger” (p.7) decrease of H3K27ac at MLL-AF4 bound enhancers vs. non-bound.
- Although a prior publication (PMID 28076791) indicated MLL-AF4 enhancers were distinct from super-enhancers, the correspondence between MLL-AF4 binding and H3K27ac is striking and suggests some of the enhancers may be classified as super-enhancers.
- A key conceptual advance is that transcription elongation machinery may be recruited by MLL-AF4 (or vice versa) at enhancers to achieve strong enhancer activity and high target gene expression. Whether a role for PAF1/FACT at enhancers is a general phenomenon or has a unique role in MLL-AF4 driven gene regulation is not clear. For example, would PAF1/FACT depletion result in loss of other TFs/regulators from enhancers due to enhancer decommissioning? Ideally, a similar PAF1 degron experiment in a non-MLLr cell line would help clarify the relationship between PAF1, MLL-FPs, and enhancer activity. Generating a new cell line is outside of the scope of requested revisions, but a PAF1 knock down strategy may be sufficient and feasible to understand how crucial PAF1 is to H3K27ac, for example, in non-MLLr cells.
- Fig.1C shows tornado plots of MLL-AF4 and H3K27ac in patient samples with the results section stating “a substantial proportion” of peaks overlap, but this proportion is not given.
- Sentence in P2 of pg. 7 is much overstated. “From this we conclude that MLL-AF4 binding generates enhancers involved in a large-scale hub of contacts (spanning tens of kb) with target promoters to activate transcription.” This is based only on correlation of patient sample MLL-AF4 binding data with capture-C in SEM cells and does not show causation. To show that MLL-AF4 binding generates enhancers, an MLL-AF4 recruitment strategy would need to be employed
- Fig. 4C axis labels missing (scale)

Response to reviewers

We'd like to thank the reviewers and the editor for their hard work assessing our original manuscript and believe that their comments have helped us significantly improve the manuscript. We are happy that they overall find our work of interest, and thank them for their generally positive comments. Reviewer 1 states: "The data are clearly presented, methodology and the presentation thereof is sound, and this report is of potential interest." Reviewer 2 also states: "In summary, I think that this manuscript is well written and the work it describes is important, impactful and technically well-done." We are also happy that Reviewer 3 thinks "The finding that MLL-AF4 is required for enhancer promoter contact is significant" and "The extensive chromatin profiling datasets generated will be a valuable resource to the broader leukemia research community."

The reviewers had many specific comments, some of which we have addressed with new data, others with new analyses and others with explanations that we hope clarify certain issues.

We hope that the changes we have made sufficiently address the reviewers' and editor's concerns. Below is our point-by-point response to the reviewers' questions and comments, with our responses in red.

Reviewer #1

Major points

1. Currently, the only non-MLL-rearranged-associated data are presented in Supplementary Figure 1h, with database-derived ATAC-seq data on various normal blood/bone marrow precursor cells for the FLT3 enhancer. Hence, no non-MLLr primary ALL-patient samples and/or ALL cell lines were investigated as a control to check for MLLr-specificity versus if some of the findings (e.g. open chromatin, H3K27ac status, etc.) are commonly found in ALL at these locations, potentially aided via wild type MLL at the same enhancer(s) in a leukemia setting. If the authors cannot access any primary non-MLLr samples, it would be good to at least see corresponding results with regards to the status of various enhancers for a couple of ALL cell lines of other subtypes than MLLr.

We'd like to thank the Reviewer for this comment as it prompted us to perform a set of new analyses which have generated interesting observations.

In order to assess whether the MLL-AF4-bound enhancers are uniquely active in *MLLr* leukemia, we made use of an H3K27ac ChIP-seq dataset generated in *MLLr* and non-*MLLr* cell lines representing a range of B cell leukemias (Kodgule et al. 2023). We found that many of these enhancers are active in multiple non-*MLLr* ALL samples, but there is a cluster of enhancers that are unique to *MLLr* leukemias (cluster 4, Supplementary Fig. 2c below). This unique cluster of highly active MLL-AF4 driven enhancers is associated with many known canonical MLL-AF4 targets (e.g. *FLT3*, *GNAQ*, *PROM1*, *HOXA7*; see Supplementary Fig. 2d below for an example at the *FLT3* locus). When we compared the expression of the genes associated with these MLL-AF4 unique enhancers using published patient RNA-seq data (Agraz-Doblas et al. 2019), we found that they are specifically highly expressed in *MLLr* patients compared to other ALLs (Supplementary Fig. 2e below, see also point 2 below).

Overall, we think that this analysis suggests that many of the MLL-AF4-bound enhancers are maintained in many ALLs, including *MLLr*. However, a key subset of these enhancers is unique to *MLLr* leukemia and driven specifically by MLL-AF4, where its binding creates a high level of functional activity and upregulation of important oncogenes. We would like to thank the Reviewer again for their comments on this, and we have modified our conclusions to reflect this nuance with the following section in the Discussion:

"Many of the MLL-AF4-bound enhancers are also active in non-*MLLr* ALL cell lines, indicating that these are common regulatory elements in leukemia. There are likely to be

distinct, MLL-AF4-independent, mechanisms maintaining these enhancers in non-*MLLr* ALL, and it is also possible that MLL-AF4 is not the main driver of enhancer activity at these loci in MLL-AF4 leukemia. In contrast, we also identified a subset of MLL-AF4-bound enhancers that were unique to *MLLr* ALL, which were associated with key oncogenes showing *MLLr*-specific transcriptional upregulation. Many of these MLL-AF4-unique enhancers were also absent from normal hematopoietic cells, indicating that they likely arise de novo during leukemogenesis. A strong example of this is the *FLT3* enhancer, which is not as large or extensive in either normal cells or other ALL samples compared to MLL-AF4 leukemias. This strongly suggests that MLL-AF4 binding may be responsible for the formation of these enhancers, however we note that from the data presented in this paper we are only able to conclude a role for MLL-AF4 enhancer maintenance. Direct targeting experiments will be required to demonstrate whether MLL-AF4 itself can generate enhancers de novo.”

Supplementary Figure 2. c, Clustered heatmap of H3K27ac levels at unified MLL-AF4-bound enhancers in MLL-AF4 patients, *MLLr* and *MLL* wild-type ALL cell lines (Kodgule et al. 2023). **e**, VST-normalized expression of genes associated with each cluster of MLL-AF4-bound enhancers indicated in (c), for *MLL* wild-type, MLL-AF4 and MLL-AF9 patient samples (Agraz-Doblas et al. 2019). * $p < 0.05$; ns: no significant difference (Mann-Whitney U test). Midline shows median, with upper and lower hinges showing 25th and 75th percentile, respectively. Upper and lower hinges extend to the largest and smallest datapoints within 1.5 times the interquartile range of either hinge.

Supplementary Figure 2. d, ATAC-seq and H3K27ac ChIP-seq at the *FLT3* and *TNFRSF14* loci in adult blood cell types (Corces et al. 2016), MLL-AF4 patients and ALL cell lines (Kodgule et al. 2023). Primary translocations are indicated. The enhancer within *PAN3* in MLL-AF4 ALL cells is highlighted in blue

2. What is the overall correlation between enhancer-occupancy by MLL::AF4 and transcriptional output from the gene predicted to be regulated by the respective enhancer? Consider using publicly available transcriptomics data on pediatric ALL for analysis of differential expression between MLL::AF4 ALL vs. non- MLL::AF4 ALL (including/excluding cases with other MLL fusion partners), with a specific focus on the overlap of MLL::AF4-associated enhancers found in the five primary ALL cases . Also, how large is that overlap ?

Within our patient RNA-seq samples, we found that genes associated with an MLL-AF4 bound enhancer are generally expressed at higher levels than non-MLL-AF4 bound enhancer associated genes (Supplementary Fig. 2b, below). Expanding beyond these samples, based on our analysis from point 1 above, we note that many of these enhancers are also active more generally in ALL. Thus, we would not expect every gene associated with an MLL-AF4-bound enhancer to be overexpressed in *MLLr* vs non-*MLLr* leukemia. Indeed, using a published RNA-seq data set (Agraz-Doblas et al. 2019), we found that genes associated with the MLL-AF4 unique enhancer set (cluster 4 from Supplementary Fig. 2c) show significantly higher expression in MLL-AF4 compared to non-*MLLr* (*MLLwt*) patient samples (Supplementary Fig. 2e, below). In addition, using several patient microarray datasets, we find that genes associated with an MLL-AF4 enhancer (cluster 4) are generally overexpressed in *MLLr* patients (Supplementary Fig. 2f, g, below). In contrast, genes associated with enhancers in other clusters (which are active in additional ALL subtypes) do not show differential expression in *MLLr* patients (Supplementary Fig. 2e, below). These new results are reported in the manuscript in the following sections:

“To assess a possible functional role for [MLL-AF4-bound] enhancers, we used a nearest-neighbor approach to assign putative enhancers to genes (Rada-Iglesias et al. 2011, Godfrey et al. 2019). In all four patients, genes associated with an MLL-AF4-bound enhancer

were more highly expressed than other genes (Supplementary Fig. 2b), implying a direct role for these enhancers in upregulating transcription.”

“To further examine the specificity of the MLL-AF4-unique enhancers, we used a published patient RNA-seq dataset (Agraz-Doblas et al. 2019) to compare the expression of enhancer-associated genes in *MLLr* and *MLLwt* leukemia. Genes associated with the MLL-AF4-unique enhancers (cluster 4) were more highly expressed in *MLLr* ALL (Supplementary Fig. 2e). In contrast, enhancers active in other ALL subtypes showed a similar level of expression in *MLLr* and *MLLwt* ALL. Using a complementary approach, we matched our 2550 MLL-AF4 enhancer gene set to 881 genes from a published ALL patient microarray dataset (Geng et al. 2012) and ranked them based on their ability to distinguish *MLLr* patient samples from other ALL subtypes. More than half of the top 50 genes were associated with the MLL-AF4-unique enhancers in cluster 4 (Supplementary Fig. 2f). Finally, using four different published patient datasets, we found that genes associated with MLL-AF4-unique enhancers were significantly overexpressed in *MLLr* patient samples compared to other ALL subtypes or normal preB cells (Supplementary Fig. 2g; ECOG E2993 (Geng et al. 2012) 84 genes up vs 36 down, $p=1.4^{-05}$; COG P9906 (Harvey et al. 2010) 114 genes up vs 16 down, $p<1^{-10}$; St. Jude 2003 (Ross et al. 2003) 83 genes up vs 28 down, $p=1.7^{-07}$; St. Jude 2013 (Figueroa et al. 2013) 70 genes up vs 14 down, $p=4.1^{-10}$). Taken together, this suggests that MLL-AF4 binding to enhancers contributes to the unique gene expression pattern observed in MLL-AF4 leukemia.”

Supplementary Figure 2. b, VST-normalized expression of genes either associated with an MLL-AF4-bound enhancer or not for the indicated patient samples. **** $p<0.0001$ (Mann-Whitney U test). Midline shows median, with upper and lower hinges showing 25th and 75th percentile, respectively. Upper and lower hinges extend to the largest and smallest datapoints within 1.5 times the interquartile range of either hinge. **e**, VST-normalized expression of genes associated with each cluster of MLL-AF4-bound enhancers indicated in (c), for MLL wild-type, MLL-AF4 and MLL-AF9 patient samples (Agraz-Doblas et al. 2019). * $p<0.05$; ns: no significant difference (Mann-Whitney U test). Midline shows median, with upper and lower hinges showing 25th and 75th percentile, respectively. Upper and lower hinges extend to the largest and smallest datapoints within 1.5 times the interquartile range of either hinge.

Supplementary Figure 2.

f, Left panel: Microarray expression analysis (ECOG E2993 (Geng et al. 2012)) of all genes associated with an MLL-AF4 enhancer. The 50 most differential genes between *MLLr* and other ALL patients are shown. Right panel: Proportion of the 50 most differential genes between *MLLr* and other ALL samples associated with each enhancer cluster indicated in (c). **g**, Microarray expression analysis of genes in four different patient datasets: i) ECOG E2993 (Geng et al. 2012); ii) COG P9906 (Harvey et al. 2010); iii) St Jude 2003 (Ross et al. 2003) and iv) St Jude 2013 (Figueroa et al. 2013). Genes used in the analysis were taken from cluster 4 (MLL-AF4 specific) in (c).

3. It is stated that gene expression data exist for 4/5 primary ALL specimens, but these data are not used within the current study. These data could, however, be used to investigate transcriptional output of genes here suggested to be regulated via the action of MLL::AF4 on the associated enhancers. Are all the respective genes (highly) expressed in these cells, i.e. is the predicted effect on transcriptional output actually seen? Linking back to both of the comments above: Albeit with a rather small sample number, a comparison between MLLr and non-MLLr cases with regards to transcriptional output of the specific genes could also be performed if the non-MLLr samples mentioned in comment #1 also were to be investigated via RNA-seq.

As mentioned in point 2 above, we followed the Reviewer's suggestion and found that, within our four *MLLr* ALL samples, MLL-AF4 enhancer-associated genes are more highly expressed than non-MLL-AF4 enhancer associated genes (Supplementary Fig. 2b, see above). Unfortunately, we were not able to access non-*MLLr* ALL patient samples for our own analysis, but instead analysed several published datasets as outlined above.

4. Western blot and/or qPCR data should be presented with regards to siRNA-mediated knockdown efficiency.

This is a very good point, and we have included western blots and ChIP-qPCR data for our previous knockdowns (Supplementary Fig. 4c and d, below), as well as for a new experiment (Supplementary Fig. 9f, below).

Supplementary Figure 4 (above). c, Western blot for AF4 (detecting MLL-AF4) and wild-type MLL in control (non-targeting; -) and 96h MLL-AF4 knockdown (+) RS4;11 and SEM cells. VINCULIN is shown as a loading control. Blots are representative of three replicates. **d,** ChIP-qPCR for MLL and AF4 at the indicated enhancer regions in SEM and RS4;11 cells, under control (NT) and MLL-AF4 knockdown conditions at 96h. Data are represented as mean \pm SEM, n=3.

Supplementary Figure 9 (left). f, Western blot for AF4 (detecting MLL-AF4), wild-type MLL, MAZ and RUNX1 in control (non-targeting; -) and 48h MLL-AF4 knockdown (+) SEM cells. VINCULIN is shown as a loading control. Blots are representative of two replicates.

Minor:

1. Should it be “Fig. 6g” on page 15, in the sentence associated with facilitation of “trans interactions between MLL-AF4 bound at the enhancer and promoter”?

We’d like to thank the Reviewer for spotting this mistake and we have corrected it in the text of the paper.

2. Please provide a full reference for the HT-ChIPmentation in the TOPmentation section in the Methods.

We have fixed this citation – this is reference 44 (Gustafsson et al. 2019) in the manuscript. Thank you for spotting this mistake.

Reviewer #2

Major points

1) Page 5: “We observed a similar distribution in a range of MLLr ALL and AML cell lines (Supplementary Fig. 1d), indicating that non-promoter binding, possibly at enhancers, is a widespread property of MLL-FPs,...”: I would suggest to formally test whether MLL-AF4 peaks show enrichment in enhancer regions.

As the Reviewer suggests, we analysed the chromatin environment of MLL-N peaks in several *MLLr* cell lines, and found that the majority of non-promoter binding sites are enriched for H3K27ac, consistent with enhancer binding (see Supplementary Figure 1f and g, below).

Supplementary Figure 1. f, Distribution of MLL ChIP-seq peaks in the indicated cell lines (or distribution of FLAG tag for FLAG-MLL-Af4 transduced cells), relative to the nearest TSS. Fusion protein expressed in each cell line is indicated. **g**, Acetylation status of MLL peaks based on distance from the nearest TSS, in the indicated *MLLr* cell lines.

We have added the following section to the results:

“While many MLL-AF4 peaks were present within 1 kb of a TSS (i.e., promoter bound), a large proportion were found more than 10 kb from the nearest TSS (Fig. 1b). We observed a similar distribution in a range of *MLLr* ALL and AML cell lines (Supplementary Fig. 1f), indicating that non-promoter binding is a widespread property of MLL-FPs. The majority of these distal MLL-FP binding sites overlapped with peaks of H3K27ac, implying binding at active enhancers (Supplementary Fig. 1g). Thus, MLL-AF4 enhancer binding is suggestive of an additional mechanism for regulating gene expression (Godfrey et al. 2017, Prange et al. 2017, Godfrey et al. 2021).”

2) Supplementary Figure 1c: It is not entirely clear how the correlation was done, i.e. does this mean number of reads in peaks? Also how was this normalized? What is the actual concordance in peaks calls? It might be a good idea to follow the standard QC procedure as suggested by the ENCODE project: <https://www.encodeproject.org/chip-seq/transcription-factor-encode4/>

We'd like to apologise for the lack of clarity in how we did this analysis. We can confirm that we followed standard QC procedures in mapping our sequencing data and generating peak files.

The figure legend has been adjusted to read the following:

"**Supplementary Figure 1. c**, Correlation of read counts from MLL ChIP-seq and TOPmentation at MLL peaks called using MLL ChIP-seq for the indicated MLL-AF4 ALL

patients. Read counts were RPKM normalized and the R^2 value was generated using the Spearman method."

We have also added the following to the "ChIP-seq/TOPmentation and ATAC sequencing analysis" section of the Methods:

"Following QC of FASTQ files by FastQC v0.12.1 (<http://www.bioinformatics.babraham.ac.uk/projects/fastqc/>), reads were trimmed using trim_galore with Cutadapt v0.6.10 (https://www.bioinformatics.babraham.ac.uk/projects/trim_galore/). Trimmed reads were then mapped to the hg19 reference genome using Bowtie 2 v2.5.1 (Langmead et al. 2009). PCR duplicates were removed using picard MarkDuplicates v3.0.0 (<http://broadinstitute.github.io/picard>). Problematic genomic regions present in the ENCODE Blacklist (<https://doi.org/10.1038/s41598-019-45839-z>) were removed from the aligned files and further QC of the aligned files was performed using samtools v1.17 (Li et al. 2009). As many of the factors that were immunoprecipitated have a mix of both sharp and broad modalities, we used either HOMER v4.11 (Heinz et al. 2010) or the deep learning based peak caller LanceOtron v1.0.8 (Hentges et al. 2021) (with a peak score cut-off value of 0.5) to call peaks. Due to the limited material available, replicates were not performed and therefore an IDR based method was not used. BigWigs were generated using the deepTools (v3.5.1) bamCoverage command (Ramirez et al. 2016), with the flags `-extendReads -normalizeUsing RPKM`, and visualized in the UCSC genome browser (Kent et al. 2002)."

3) Page 5: "We saw a strong correlation between levels of H3K27ac and MLL-AF4 binding at both promoters and enhancers (Fig. 1c, e), indicating that MLL-AF4 associates with enhancer as well as promoter activity in primary ALL cells.": Why is the correlation between MLL binding and H3K27ac weaker at enhancers than at promoters? Is this expected?

This is an interesting observation. We believe this is because many enhancers in these cells are not bound by MLL-AF4 (see Supplementary Fig. 1i, below), and at these enhancers there will be low levels of MLL but, potentially, high levels of H3K27ac. In contrast, MLL-AF4 is bound at a much larger proportion of gene promoters (Supplementary Fig. 1d, below; and see (Kerry et al. 2017)), so the correlation with H3K27ac is more consistently maintained. In support of this explanation, if we limit the comparison to just MLL-AF4-bound enhancers, the correlation between MLL and H3K27ac is comparable to what we see at promoters (Fig. 1a, below).

Supplementary Figure 1. i, Proportion of MLL-AF4-bound and non-MLL-AF4-bound enhancers located within (intragenic) or between (intergenic) genes in SEM cells. **d**, Overlap of genes bound by MLL-AF4 at promoters in the indicated MLL-AF4 ALL patients.

Figure 1.
a, Correlation of H3K27ac and MLL ChIP-seq/TOPmentation signal at all ATAC peaks, promoter ATAC peaks, all enhancers and MLL-AF4-bound enhancers, for the indicated patient samples.

4) Page 6: “This increased accessibility appears to be associated with leukemogenesis, as, for example, the high density of ATAC-seq peaks observed at the FLT3 enhancer in MLL-AF4 patients and cell lines is completely absent in normal cells, including those from the B lineage (Supplementary Fig. 1h)”: I would suggest to support this statement by a more systematic (genome-wide) analysis, e.g. by showing the read coverage at MLL-enhancers. Similarly, I suggest to support the statement related to the MCC interactions which relate to Fig 1g with a more systematic, genome-wide analysis.

We’d like to thank the Reviewer for bringing up this point, as the analysis we conducted to address this question, combined with point 1 from Reviewer 1, has revealed a more nuanced picture of MLL-AF4-bound enhancer regulation. Unfortunately, because Micro-Capture-C (MCC) is a capture 3C method (meaning that it is targeted to specific loci), the trade-off for high resolution is that it does not generate genome-wide data for analysis (see also our response to point 5, below).

Nonetheless, we took the Reviewer’s suggestion of broadening our analysis by comparing ATAC-seq data at MLL-AF4 enhancers genome-wide. To do this, we compared the density of ATAC-seq signal between our MLL-AF4 primary samples and normal hematopoietic cells. We found that many of these enhancers are active in normal cells, but there is a specific cluster of enhancers at which the ATAC-seq signal is more enriched in *MLLr* samples than in normal cells (cluster 3, Supplementary Fig. 3f, below). One example from this cluster is represented by the enhancer we previously highlighted at the *FLT3* locus (Supplementary Fig. 2d, below). As well as having a broad domain of open chromatin, this region also has high levels of H3K27ac compared to other ALL samples (Supplementary Fig. 2d, below).

Supplementary Figure 3.
f, Clustered heatmap of ATAC-seq levels at unified MLL-AF4-bound enhancers in MLL-AF4 patients and adult blood cell types (Corces et al. 2016).

Supplementary Figure 2. d, ATAC-seq and H3K27ac ChIP-seq at the *FLT3* and *TNFRSF14* loci in adult blood cell types (Corces et al. 2016), MLL-AF4 patients and ALL cell lines (Kodgule et al. 2023), MLL-AF4 patients. Primary translocations are indicated. The enhancer within *PAN3* in MLL-AF4 ALL cells is highlighted in blue.

When we do a similar analysis to compare H3K27ac signal, as a measure of enhancer activity, between *MLLr* and non-*MLLr* cell lines representing a range of B cell leukemias, we find many of these enhancers are active in multiple ALL samples, but there is a cluster of enhancers that are more highly active and more highly expressed in *MLLr* leukemias (cluster 4, Supplementary Fig. 2c and e, below). Many of these highly active MLL-AF4 enhancers are associated with known canonical MLL-AF4 targets (e.g. *FLT3*, *GNAQ*, *PROM1*, *ARID1B*, *CDK6* and *MYC*), and we have included several examples in the manuscript to illustrate that these enhancers display broad interaction domains as detected by MCC (Fig. 1g, Supplementary Fig. 3d, below). Unfortunately, as we are not able to perform MCC genome-wide we don't know how common these MLL-AF4 high density enhancers actually are, but our ATAC-seq analysis suggests that there is a cluster of MLL-AF4 enhancers that can display this activity. To reflect these nuances, we have added these points into the Results in the following sections:

“Visual inspection of ATAC-seq data at key broad MLL-AF4-bound enhancers, for example at the *FLT3*, *ARID1B*, *CDK6* and *MYC* enhancer loci (Fig. 1g, Supplementary Fig. 3d), revealed a high density of peaks of open chromatin, especially when compared to other enhancers, such as at *LMO4*, *IKZF3* and *SMAD3* (Fig. 1g, Supplementary Fig. 3e), suggesting a high frequency of protein binding. This increased accessibility appears to be associated with leukemogenesis, as, for example, the high density of ATAC-seq peaks observed at the *FLT3* enhancer in MLL-AF4 patients and cell lines is completely absent in normal cells, including those from the B lineage (Supplementary Fig. 2d) (Corces et al. 2016, O'Byrne et al. 2019). We explored whether MLL-AF4 associates with highly accessible enhancers genome-wide by comparing ATAC-seq signal enrichment at the 2550 MLL-AF4-bound enhancer set (Supplementary Table 2) with published ATAC-seq from normal hematopoietic cells (Corces et al. 2016). While some of these enhancers displayed highly

enriched signal specifically in the MLL-AF4 patient blasts (Supplementary Fig. 3f; cluster 3), a large proportion also showed high levels of accessibility in common lymphoid progenitor (CLP) cells (clusters 4 and 5). Other MLL-AF4 enhancers, for example at *TNFRSF14* (Supplementary Fig. 2d), were more active in non-B lineage cell types (clusters 1 and 2). The MLL-AF4 specific ATAC cluster (cluster 3) contained such canonical genes as *FLT3* (Supplementary Fig. 2d), *PROM1*, *RUNX2*, *ARID1B*, *MBNL1* and *JMJD1C* (Supplementary Table 2).”

“A key feature of most enhancers is a high frequency of interactions with target gene promoters. To explore this at MLL-AF4-bound enhancers, we used the ultra-high resolution technique Micro-Capture-C (MCC) (Hua et al. 2021) to look at a subset of key oncogenes. Although MCC does not allow genome-wide interaction analysis, the high sequencing depth permits precise mapping of DNA-DNA interactions at single base pair resolution (Hua et al. 2021). The MCC interaction profiles for *FLT3*, *ARID1B*, *CDK6* and *MYC* are markedly broad, showing extensive interactions aligning with the high density of ATAC-seq peaks (likely TF binding events) (Hua et al. 2021) (Fig. 1g, left, Supplementary Fig. 3d). These interactions also broadly correlate with MLL-AF4 binding, suggesting that regions densely bound by MLL-AF4 directly contact the promoter (Fig. 1g, Supplementary Fig. 3d). “

“Since MCC is a not a genome-wide technique, we were not able to verify what proportion of MLL-AF4 bound enhancers display these high-density interaction profiles. However, we conclude that, at a subset of highly active enhancers, MLL-AF4 binding is associated with a large-scale hub of contacts (spanning tens of kb) with target promoters to activate transcription.”

Supplementary Figure 2. c, Clustered heatmap of H3K27ac levels at unified MLL-AF4-bound enhancers in MLL-AF4 patients, *MLLr* and *MLL* wild-type ALL cell lines (Kodgule et al. 2023). **e**, VST-normalized expression of genes associated with each cluster of MLL-AF4-bound enhancers indicated in (c), for MLL wild-type, MLL-AF4 and MLL-AF9 patient samples (Agraz-Doblas et al. 2019). * $p < 0.05$; ns: no significant difference (Mann-Whitney U test). Midline shows median, with upper and lower hinges showing 25th and 75th percentile, respectively. Upper and lower hinges extend to the largest and smallest datapoints within 1.5 times the interquartile range of either hinge.

Figure 1. g, Capture-C, Micro-Capture-C (MCC), ATAC-seq and ChIP-seq for MLL, AF4 and H3K27ac at the *FLT3* and *LMO4* loci in SEM cells. Enhancer regions are highlighted in purple. Capture-C and MCC traces scaled to emphasize distal interactions.

Supplementary Figure 3. d, Capture-C, Micro-Capture-C (MCC), ATAC-seq and ChIP-seq for MLL, AF4 and H3K27ac at the *ARID1B*, *CDK6* and *MYC* enhancer loci in SEM cells. Enhancer regions are highlighted in purple. Capture-C and MCC traces scaled to emphasize distal interactions.

5) Page 6: “We intersected SEM enhancers and MLL-AF4 peaks, identifying enhancers enriched (bound) or depleted (not bound) for MLL-AF4 (Fig. 2a), and linked these to the nearest gene.”: Why were the enhancers linked to the closest gene? The authors have data for gene promoters interacting with the individual enhancers. Why didn’t they use this information? I suggest that this should be done in a systematic, data-driven way.

The trade off with 3C capture methods such as Capture C (Davies et al. 2016) and Micro Capture C (Hua et al. 2021) is that they provide very high resolution for enhancer-promoter interactions, but the sequencing depth required precludes analysis of more than a few dozen target genes without prohibitive cost, so the data are not genome-wide. Unfortunately, therefore, we only have data for enhancer-promoter interactions at a subset of targets. The large number of high-quality cells needed for a sufficiently complex 3C library limits this technique to cell lines, so we were not able to generate equivalent data in primary patient

samples. Because of these constraints, we chose to use the imperfect “nearest gene approach” to identify enhancer-gene regulations. This is an approach that has been used successfully to identify developmental enhancers (Rada-Iglesias et al. 2011) and in cases where we were able to compare this to our Capture-C/MCC data, it accurately links many important enhancers to their target genes. We do however agree with the Reviewer that in future, it would be better to have an experimentally-validated interaction profile, perhaps by scaling up MCC or using lower resolution but genome wide approaches such as Micro C (Hsieh et al. 2015).

6) Page 12: “Together this suggests that broad MLL-AF4 binding domains are associated with increased TF binding, indicating that MLL-AF4 may promote TF binding to contribute to aberrant enhancer activation.”: To confirm this finding, I suggest that the authors perform RUNX1 and MAZ1 ChIP-seq after MLL-AF4 KD. If MLL-AF4 is the driver for TF binding events as the presented data suggest, then TF peaks should disappear upon MLL-AF4 KD.

To answer this point, we performed MLL-AF4 knockdown for 48 hours (Supplementary Fig. 9f, below). At this time point, MAZ protein levels are unaffected and RUNX1 protein levels are very slightly reduced (Supplementary Fig. 9f, below), likely because of MLL-AF4 regulation of the *RUNX1* locus (Wilkinson et al. 2013, Harman et al. 2021). We observed a reduction in RUNX1 binding with the loss of MLL-AF4 binding (Fig. 6f and Supplementary Fig. 9g, below), but, surprisingly, MAZ binding actually increased (Supplementary Fig. 9g and h, below). We are not sure exactly what this means, but it suggests that the relationship between MLL-AF4 and TF binding at MLL-AF4 enhancers may vary between specific TFs. RUNX1 binding appears to be dependent on MLL-AF4, suggesting a more straightforward interaction in which MLL-AF4 spreading may promote RUNX1 association. For MAZ, however, the effect we observed could be based on the fact that its zinc finger domain binds tightly to CpG sequences (Ashfield et al. 1994) (potentially competing with the CXXC domain of MLL-AF4), and thus binds more frequently in the absence of MLL-AF4. MLL-AF4-bound regions tend to be hypomethylated, which may explain the high frequency of MAZ binding in the steady state (Kerry et al. 2017). To reflect this new data, we have added the following section to the results:

“Together these analyses suggest that broad MLL-AF4 binding domains are associated with increased TF binding, indicating that MLL-AF4 may help maintain TF binding to contribute to aberrant enhancer activation. We tested this dependency by conducting ChIP-seq for RUNX1 and MAZ following MLL-AF4 knockdown. As RUNX1 and MAZ are both positively regulated by MLL-AF4 (Harman et al. 2021), we chose a 48h time-point, to minimize impacts on RUNX1 and MAZ protein levels due to gene expression changes. At this time point, MAZ protein levels remained stable, but RUNX1 protein levels were very slightly reduced (Supplementary Fig. 9f). MLL-AF4 KD resulted in a reduction of RUNX1 binding at broad MLL-AF4 enhancers (Fig. 6f), for example at *ARID1B* (Supplementary Fig. 9g), suggesting that RUNX1 binding at these loci dependent on MLL-AF4. In contrast, surprisingly, MAZ binding increased at enhancers following MLL-AF4 KD, suggesting potential competitive binding between MAZ and MLL-AF4 (Supplementary Fig. 9h). One possibility is that hypomethylation of MLL-AF4-bound regions (Kerry et al. 2017) partly contributes to increased DNA binding of factors such as MAZ, which generally have a preference for CG rich DNA (Ashfield et al. 1994). In the absence of MLL-AF4, DNA methylation is unlikely to be rapidly re-established, meaning that, without competition, MAZ binding at these loci may increase. Thus, there may be a complex interplay between MLL-AF4 complex activity and DNA hypomethylation driving binding of some TFs at MLL-AF4 enhancers.”

Supplementary Figure 9 f, Western blot for AF4 (detecting MLL-AF4), wild-type MLL, MAZ and RUNX1 in control (non-targeting; -) and 48h MLL-AF4 knockdown (+) SEM cells. VINCULIN is shown as a loading control. Blots are representative of two replicates. **g**, ChIP-seq for MLL, RUNX1 and MAZ at *ARID1B* under control (NT) and 48h MLL-AF4 knockdown conditions. Putative enhancers are highlighted. **h**, Mean MAZ ChIP-seq signal under control (NT) and 48h MLL-AF4 knockdown conditions, at expressed promoters of genes containing an MLL-AF4 binding domain >50 kb, over a 6 kb (*left*) or 80 kb (*right*) window.

Figure 6 f, Mean ChIP-seq signal under control (NT) and 48h MLL-AF4 knockdown conditions, at expressed promoters of genes containing an MLL-AF4 binding domain >50 kb, over a 6 kb (*left*) or 80 kb (*right*) window.

Reviewer #3

Major points

1) Key experimental details are missing and should be added to the current manuscript even if reported in the authors' prior publications, including data on how well the MLL-AF4 knockdown worked (western blot), and the time post-perturbation (siRNA transfection or dTAG13 treatment) of conducting RNA-seq/ChIP-seq/Capture-C. The only indication is a reference to their prior study for MLL-AF4 knock-down, where they reported waiting 24 hours before conducting an assay post knock-down. Is that the same here? It seems like it may be too short for protein loss due to RNA knockdown to impact global chromatin structure changes reported here, especially as the authors acknowledge that differences in treatment time may underlie conflicting results observed in this study and a prior study (ref80 10.1016/j.molcel.2021.05.028) Clarifying the critical temporal component of each experiment is highly recommended.

We'd like to thank the Reviewer for pointing out this oversight. We've now added details of each experiment to the figure legends and Methods section. Most MLL-AF4 siRNA knockdowns were done twice over 96 hours (2x48 hours), with samples collected at 96 hours, except for the data in Supplementary Fig. 9, where cells were harvested after 48 hours. Degrons were analysed after a 24 hour treatment. Western blots and ChIP-qPCR/ChIP-seq experiments validating the MLL-AF4 knockdowns (Supplementary Fig. 4c and d, Supplementary Fig. 9f) and PAF1 and SSRP1 degron experiments (Fig. 4a, Supplementary Fig. 7c) are now included.

Figure 4.
a, Western blot for PAF1 or SSRP1 in wild-type (WT), PAF1 degron or SSRP1 degron cells, with (+) or without (-) addition of 0.5 μM dTAG-13 for 24h. Blots are representative of three replicates. Bands representing wild-type and FKBP12^{F36V}-tagged proteins are indicated.

Supplementary Figure 7.
c, Western blot for PAF1 in wild-type (WT) and a pool of PAF1 degron MM1.S cells, with (+) or without (-) addition of 0.5 μM dTAG-13 for 24h. Blots are representative of three replicates. Bands representing wild-type and FKBP12^{F36V}-tagged PAF1 are indicated.

2) There are several points of contradiction with both the authors' and others' prior work, most notably regarding the role of PAF1. This is not adequately addressed in the discussion, especially the study by Chen et al (PMID 28860207) which showed that PAF1 loss is an important step in enhancer activation using human cancer cell lines, in contrast to this current study where the authors show that PAF1 loss leads to enhancer decommissioning. Prior literature also showed that PAF1 regulates PolII pause release, and PAF1 depletion led to increased transcription due to release from paused Pol II therefore the results in the current study are confusing as they see that PAF1 depletion leads to a reduction in transcription. The authors discuss these contradictions only superficially and conclude that the differences may simply be due to a difference in treatment time (again highlighting the need to clarify each treatment design in this current study).

When the reviewer mentions contradictions with our own previous work, we think they are referring to the fact that using the TetR system, in a previous paper, we failed to detect an interaction between full-length MLL-AF4 and PAF1 (Kerry et al. 2017). We think this is because PAF1 interacts weakly with the CXXC domain. We were never able to make a stable cell line expressing TetR-MLL-AF4, owing to the toxicity of the protein, and the low level of expression by transient transfection likely contributed to the challenge in detecting a PAF1 interaction. In the current manuscript, we applied a different strategy, expressing the CXXC domain alone in a stable line, allowing for higher expression. With this approach we were able to successfully detect a PAF1 interaction with the CXXC domain. We have explained this in our results:

"We generated mouse embryonic stem cell lines expressing individual components of the MLL-AF4 complex fused to TetR, which binds at an array of TetO repeats inserted into the mouse genome (Blackledge et al. 2014). Previously, we have used transient transfection to express low levels of TetR-MLL-AF4, owing to protein toxicity, but we were unable to detect the interaction between MLL and PAF1C, possibly because it is a weaker interaction that that observed with core complex members such as MENIN (Kerry et al. 2017). To increase the sensitivity of the assay, we created a stable cell line expressing the MLL CXXC domain alone (TetR-CXXC), as well as TetR fusions of other complex components. Using CHIP-

qPCR to assess the ability of these proteins to recruit other factors to the *TetO* array, we detected *in vivo* interactions between TetR-CXXC with PAF1 and SSRP1 (Fig. 3k).”

We also agree with the Reviewer that we did not properly address the role of PAF1 in gene regulation. We have added new work targeting PAF1 for degradation in the multiple myeloma cell line, MM1.S, to provide a comparison to the SEM (MLL-AF4) cells (discussed in more detail below, in response to minor point 4). Briefly, we found that PAF1 is also bound at enhancers in MM1.S cells, although PAF1 degradation did not strongly affect enhancer activity (H3K27ac or eRNA levels) – importantly, we did not see an increase in enhancer features. However, we did observe reduced transcriptional elongation at genes, indicating that PAF1 has a positive transcriptional role in both SEM and MM1.S cells. The fact that we observe distinct effects of PAF1 degradation on enhancer loss in SEM and MM1.S cells indicates that the contradiction of our findings with the Chen et al study (Chen et al. 2017) may be a consequence of the different cell types used.

In addition to these new experiments, we have extensively revisited the role of PAF1 in the discussion with the following sections:

“While PAF1 was originally identified as a transcriptional elongation factor, it has since been found to regulate multiple stages of RNAPII transcription (Van Oss et al. 2017, Francette et al. 2021). Perhaps because of this pleiotropy, *in vivo* studies of PAF1C function have reported contradictory effects of PAF1 depletion on gene expression. In some, knockdown of PAF1 increased RNAPII and super elongation complex (SEC) occupancy across gene bodies, suggesting a role in suppressing promoter-proximal pause release (Bai et al. 2010, Chen et al. 2015). However, other studies instead showed that loss of PAF1C impaired transcription elongation, indicating that PAF1 was essential for productive promoter-proximal pause release (Wu et al. 2014, Yu et al. 2015, Lu et al. 2016, Hou et al. 2019, Zumer et al. 2021), consistent with *in vitro* transcription and structural studies (Kim et al. 2010, Vos et al. 2018, Farnung et al. 2022). It is possible that PAF1C has different roles at different genes or in different cellular contexts; our results here are consistent with PAF1C promoting transcription elongation in both MLL-AF4 ALL and multiple myeloma cells.

“Recently, use of degron technology has allowed the consequences of acute PAF1C degradation to be investigated. One study in K562 (chronic myelogenous leukemia) cells identified distinct requirements for different PAF1C components in stimulating RNAPII activity, with degradation of the PAF1 subunit only weakly affecting RNA synthesis (Zumer et al. 2021). This is in contrast to the dramatic reduction in transcription produced by PAF1 degradation in SEM (MLL-AF4 ALL) cells, but matches the much weaker effect we observed in MM1.S (multiple myeloma) cells. Thus, while PAF1 may have a particularly central role in transcription in MLL-AF4 leukemias, this is not universal, and may be a consequence of the activity and multivalent interactions of the MLL-AF4 complex. A full answer to this question will be revealed through studying PAF1C function in a wider range of cell types and tissues.

“While relatively understudied, roles for PAF1C at enhancers have also been proposed (Chen et al. 2017, Ding et al. 2021, Liu et al. 2022). Our finding that loss of PAF1C reduced enhancer activity, not only at MLL-AF4 bound enhancers, but also at many non-MLL-AF4 bound enhancers, complements a recent study showing a strong correlation between PAF1C binding and enhancer activity (Ding et al. 2021), arguing for a role in promoting transcription. In contrast, Chen et al (Chen et al. 2017) proposed that PAF1C restrains activation of a subset of enhancers by inhibiting RNAPII promoter-proximal pause release. Another study found that PAF1 knockout resulted in increased eRNA levels, which the authors attributed to PAF1 recruitment of Integrator to terminate transcription (Liu et al. 2022). In our work we observe the opposite effect; PAF1 degradation significantly decreased eRNA transcription in SEM cells. One possibility is that the timing of PAF1 depletion is critical; our assays were conducted after 24h, whereas both previous studies used a window of 72h or longer (Chen et al. 2017, Liu et al. 2022), which could allow for additional secondary effects on enhancer activity. Surprisingly, while we observed a strong effect of

PAF1 depletion on enhancer function in SEM cells, the consequences were minimal on MM1.S enhancer function, despite the high frequency of PAF1 binding at enhancers. This may also explain why SEM cells displayed a clear reduction in transcription initiation, whereas MM1.S cells only showed an effect on elongation. Together, these observations support the notion that PAF1C activity may vary based on the cell type, perhaps dependent on the presence or absence of specific complex components, or the chromatin context itself.”

Minor points

1) The Reviewer suggested several analyses for the MLL-AF4 patient data. We’d like to thank them for these suggestions as we feel they have provided additional context to characterising MLL-AF4 behavior in these samples.

MLL-AF4 binding in patient samples is an important technical achievement and the authors state that enhancer binding may help determine the biology of the leukemias. The patient data seem under-utilized, especially the observation of putative MLL-AF4-driven de novo enhancers (e.g. in PAN3). How many promoters are bound by MLL-AF4 in primary leukemia cells and are they all active?

This analysis is now included in Supplementary Fig. 1d and e. There is a high level of overlap in promoter binding between the five samples for which we have MLL-AF4 ChIP-seq. Approximately 60% of MLL-AF4 bound genes (6021) were common between all patients, and very few promoters (≤ 184) were bound in only one sample (Supplementary Fig. 1d, below). Consistent with its role in promoting gene expression, approximately 80% of bound genes were transcriptionally active (Supplementary Fig. 1e, below). This new analysis has been added to the results section.

Supplementary Figure 1. d, Overlap of genes bound by MLL-AF4 at promoters in the indicated MLL-AF4 ALL patients. **e**, Proportion of MLL-AF4-bound gene promoters that are expressed in each patient.

What is the correspondence between primary cell binding and SEM binding?

MLL-AF4 enhancer binding in SEM cells is highly correlated (in most cases $R \sim 0.8$) with enhancer binding in patient samples. This new data has been included as Supplementary Fig. 1j (below).

Supplementary Figure 1. j, Correlation of read counts from MLL ChIP-seq from MLL-AF4 patients and SEM cells at the unified MLL-AF4-bound enhancer set.

How many putative de novo enhancers were found?

We compared our analysis across the four patient lines for which we have H3K27ac and MLL-AF4 binding data, and found 2550 enhancers bound by MLL-AF4 in at least three of the four patients. This data is now included as Supplementary Fig. 2a (below).

Supplementary Figure 2. a, Overlap of enhancer usage between MLL-AF4 patients. Enhancers present in the unified MLL-AF4-bound enhancer set (bound in at least three patients) are colored purple.

What is the correspondence between the 5 patients?

As described above, MLL-AF4 binding at promoters (Supplementary Fig. 1d) and ChIP-seq signal at enhancers (Supplementary Fig. 1j, Supplementary Fig. 2a) was highly correlated between the patient samples.

Which TF motifs are enriched in the ATAC-seq peaks of leukemia-specific MLL-AF4 bound enhancers?

This was an interesting suggestion from the Reviewer. We looked for motif enrichment at ATAC-seq peaks found within MLL-AF4-bound enhancers, but overall did not find anything which we felt was biologically meaningful in explaining MLL-AF4 binding or enhancer activity. This is included in the results with the following passage:

“We used motif enrichment to identify whether specific TFs were associated with MLL-AF4-bound enhancers. While several TF motifs were found to be statistically enriched at MLL-AF4-bound enhancers, including for MEF2D, ATF3 and CREB1, the proportion of enhancers containing these motifs was broadly similar globally at all enhancers, suggesting that MLL-AF4-bound enhancers are unlikely to be defined by specific TF binding sequences (Supplementary Fig. 9a).”

Supplementary Figure 9.
a, Enrichment of transcription factor motifs at all enhancers and the unified MLL-AF4-bound enhancer set in MLL-AF4 patients.

What are their presumed target genes?

Following a suggestion from Reviewer #1, we compared H3K27ac ChIP-seq (as a mark of enhancer activity) at the patient MLL-AF4-bound enhancers between *MLLr* and non-*MLLr* cell lines representing a range of B cell leukemias. We found that many MLL-AF4 bound enhancers are active in multiple ALL samples, but there is a cluster of enhancers that are specifically enriched in *MLLr* leukemias (cluster 4; Supplementary Fig. 2c, below).

In order to ask which genes these enhancers regulate, we used the common approach of linking them to the nearest gene promoter. We found that this cluster of highly active MLL-AF4 driven enhancers is associated with many known canonical MLL-AF4 targets (e.g., *FLT3*; Supplementary Fig. 2d, below; see also Supplementary Table 2) and that these genes are specifically highly expressed in *MLLr* patients compared to other ALLs (Supplementary Fig. 2e, below). This suggests to us that many known MLL-AF4 target genes are driven partly through MLL-AF4-bound enhancer activity.

We have a long section reporting this new analysis in the data, but the essence is contained in this section:

“We asked whether the 2550 enhancers bound by MLL-AF4 were unique to MLL-AF4 ALL, by comparing H3K27ac levels at these enhancers, as a proxy for activity, in cell lines representing different ALL subtypes (Supplementary Fig. 2c and Supplementary Table 2) (Kodgule et al. 2023). As expected, *MLLr* cell lines clustered with MLL-AF4 patients and showed unique activity at a subset of MLL-AF4-bound enhancers (cluster 4), for example at the *FLT3* enhancer (Supplementary Fig. 2d). Genes that are associated with these enhancers include canonical MLL-AF4 targets, such as *MEIS1*, *FLT3*, *PROM1*, *HOXA7*,

CCNA, *CPEB2*, *RUNX2*, *ARID1B*, *MBNL1* and *JMJD1C* (Armstrong et al. 2002, Kerry et al. 2017, Godfrey et al. 2019, Godfrey et al. 2021) (Supplementary Table 2).”

Supplementary Figure 2. c, Clustered heatmap of H3K27ac levels at unified MLL-AF4-bound enhancers in MLL-AF4 patients, *MLLr* and *MLL* wild-type ALL cell lines (Kodgule et al. 2023). **e**, VST-normalized expression of genes associated with each cluster of MLL-AF4-bound enhancers indicated in (c), for MLL wild-type, MLL-AF4 and MLL-AF9 patient samples (Agraz-Doblas et al. 2019). * $p < 0.05$; ns: no significant difference (Mann-Whitney U test). Midline shows median, with upper and lower hinges showing 25th and 75th percentile, respectively. Upper and lower hinges extend to the largest and smallest datapoints within 1.5 times the interquartile range of either hinge.

Supplementary Figure 2. d, ATAC-seq and H3K27ac ChIP-seq at the *FLT3* and *TNFRSF14* loci in adult blood cell types (Corces et al. 2016), MLL-AF4 patients and ALL cell lines (Kodgule et al. 2023), MLL-AF4 patients. Primary translocations are indicated. The enhancer within *PAN3* in MLL-AF4 ALL cells is highlighted in blue.

2) The plots used to show enrichment and/or change in read counts for different factors are good summaries (“mean distribution”) but may oversimplify the data (it’s not possible to gauge the distribution of fold changes observed at all peaks by reducing the data to a single line showing the mean). At least for a few key figures (2c, 4d), having a volcano plot or violin/box plot that shows the fold change of reads for each peak analyzed (i.e. all show all data points) may help better show relationship and could point to sets of enhancers that are more sensitive to MLL-AF4 loss (in terms of PAF1 binding or histone mod) than others. For example, in Figure 2c it is claimed that MLL-AF4 bound enhancers show a greater loss of H3K27ac compared to non-MLL-AF4 bound enhancers following MLL-AF4 knockdown, however the non-MLL-AF4 bound enhancers have a lower H3K27ac signal to begin with; thus a more fair visual display would indicate differences in fold-changes rather than total read counts. This would also clarify whether there is truly a “much larger” (p.7) decrease of H3K27ac at MLL-AF4 bound enhancers vs. non-bound.

We’d like to thank the Reviewer for these suggestions. The data from Fig. 2c are now also displayed as a violin plot in Supplementary Fig. 4e (below). We note that, as the Reviewer indicated, there is a decrease in H3K27ac at all enhancers, although there is a statistically significant difference, if subtle, between the change at MLL-AF4-bound and -unbound enhancers.

Supplementary Figure 4. e, Log2 fold-change in levels of H3K27ac, H3K79me3, PAF1 and RNA transcription at enhancers following 96h MLL-AF4 knockdown in SEM cells. Statistical significance calculated using a Wilcoxon rank sum test. Midline shows median, with upper and lower hinges showing 25th and 75th percentile, respectively. Upper and lower hinges extend to the largest and smallest datapoints within 1.5 times the interquartile range of either hinge.

The data in Figure 4b and d are also displayed as a violin plot and a scatter plot in Supplementary Fig. 6d and g, respectively.

Supplementary Figure 6. d, Log2 fold-change in levels of RNA transcription at enhancers following 24h dTAG-13 treatment of PAF1 degnon or SSRP1 degnon SEM cell lines. Statistical significance calculated using a Mann–Whitney U test. Midline shows median, with upper and lower hinges showing 25th and 75th percentile, respectively. Upper and lower hinges extend to the largest and smallest datapoints within 1.5 times the interquartile range of either hinge. **g**, Levels of H3K27ac, H3K79me3 and MLL at MLL-AF4-bound (purple) and non-MLL-AF4-bound (gray) enhancers following 24h dTAG-13 treatment of PAF1 degnon or SSRP1 degnon SEM cell lines.

3) Although a prior publication (PMID 28076791) indicated MLL-AF4 enhancers were distinct from super-enhancers, the correspondence between MLL-AF4 binding and H3K27ac is striking and suggests some of the enhancers may be classified as super-enhancers.

This is an excellent point from the Reviewer. Although we classify “spreading targets” (PMID 28076791 (Kerry et al. 2017)) and MLL-AF4-bound enhancers (this study) using different analytical approaches, there is indeed an overlap between the two, and some of these MLL-AF4 enhancers are super-enhancers. We have included this analysis in Supplementary Fig. 3b and c (below). About 50% of MLL-AF4-bound enhancers can be classified as super-enhancers. We’d like to thank the reviewer for pointing this out.

Supplementary Figure 3. b, Proportion of MLL-AF4-bound and unbound enhancers in SEM cells that are classified as super-enhancers. **c**, Distribution of the length of super- and typical enhancers bound or not bound by MLL-AF4 in SEM cells. Statistical significance calculated using a Mann-Whitney U test. Midline shows median, with upper and lower hinges showing 25th and 75th percentile, respectively. Upper and lower hinges extend to the largest and smallest datapoints within 1.5 times the interquartile range of either hinge.

4) A key conceptual advance is that transcription elongation machinery may be recruited by MLL-AF4 (or vice versa) at enhancers to achieve strong enhancer activity and high target gene expression. Whether a role for PAF1/FACT at enhancers is a general phenomenon or has a unique role in MLL-AF4 driven gene regulation is not clear. For example, would PAF1/FACT depletion result in loss of other TFs/regulators from enhancers due to enhancer decommissioning? Ideally, a similar PAF1 degron experiment in a non-MLLr cell line would help clarify the relationship between PAF1, MLL-FPs, and enhancer activity. Generating a new cell line is outside of the scope of requested revisions, but a PAF1 knock down strategy may be sufficient and feasible to understand how crucial PAF1 is to H3K27ac, for example, in non-MLLr cells.

In our experience, it is very difficult to get a PAF1 (or FACT) knockdown that is efficient enough to produce a transcriptional phenotype, as the small amount of residual protein appears to be sufficient for activity. Therefore, in order to address the Reviewer’s question, it was necessary to generate a novel PAF1 degron. We chose to do this in MM1.S cells, which represent multiple myeloma, a B lineage malignancy unrelated to MLL-AF4. We found that PAF1 is also enriched at enhancers in these cells, as in MLL-AF4 cells (Supplementary Fig. 7a and b, below). Owing to time constraints, we were unable to generate a clonal population of homozygous PAF1 degron-tagged cells, instead using a pool of (mNeonGreen-positive) homozygotes and heterozygotes (Supplementary Fig. 7c). Despite this, degradation of PAF1

clearly reduces transcription elongation (Supplementary Fig. 7d) and has a subtle effect on eRNA transcription (Supplementary Fig. 7e) but little effect on enhancer H3K27ac levels (Supplementary Fig. 7f and g). We note that in SEM cells, PAF1 degradation had a dramatic effect on both transcription initiation and elongation (Supplementary Fig. 6c, below), whereas in MM1.S cells we only observed an effect on the elongation phase (Supplementary Fig. 7d), which may point to a role in transcription initiation in SEM cells. In all, the effect of PAF1 degradation on enhancer function is much reduced in multiple myeloma cells compared to SEM cells. It is possible that this is due to partial PAF1 degradation in this pool of cells, but it may also suggest a more unique role for PAF1 in enhancer function in MLL-AF4 leukemias.

Supplementary Figure 7.

a, Proportion of super- (SE) and typical (TE) enhancers bound by PAF1 in MM1.S cells. **b**, PAF1 ChIP-seq, reference-normalized H3K27ac ChIP-seq and TT-seq at the *CCND2* locus in PAF1 degron MM1.S cells, with or without the addition of dTAG-13 for 24h. Enhancers are highlighted in blue. **c**, Western blot for PAF1 in wild-type (WT) and a pool of PAF1 degron MM1.S cells, with (+) or without (-) addition of 0.5 μ M dTAG-13 for 24h. Blots are representative of three replicates. Bands representing wild-type and FKBP12^{F36V}-tagged PAF1 are indicated. **d**, Metagene profiles of TT-seq levels across gene bodies in PAF1 degron MM1.S cells under control (untreated) and 24h dTAG-13-treated conditions, stratified into quartiles by gene length. **e**, Mean distribution of strand-specific TT-seq (eRNA) levels at inter- and intragenic enhancers, in PAF1 degron MM1.S cells under control (untreated) and 24h dTAG-13-treated conditions. **f**, Mean distribution of H3K27ac at inter- and intragenic enhancers, in PAF1 degron MM1.S cells under control (untreated) and 24h dTAG-13-treated conditions. **g**, H3K27ac levels at PAF1-bound and -unbound enhancers, in PAF1 degron MM1.S cells under control (untreated) and 24h dTAG-13-treated conditions.

Supplementary Figure 6. c, Metagene profiles of TT-seq levels across gene bodies in PAF1 degran or SSRP1 degran cell lines, stratified into quartiles by gene length.

5) Fig.1C shows tornado plots of MLL-AF4 and H3K27ac in patient samples with the results section stating “a substantial proportion” of peaks overlap, but this proportion is not given.

We have quantified this in several ways. Firstly, we looked at the chromatin environment of MLL peaks within each patient. We found that approx. 30-50% of MLL peaks are at promoters, and in each patient approx. 20-25% of MLL peaks overlap with non-promoter H3K27ac peaks, i.e., putative enhancers (Fig. 1d, below).

Figure 1. d, Proportion of MLL peaks associated with promoters and enhancers in each patient sample.

In addition, we looked at common and unique enhancers in the four patient samples for which we have H3K27ac data, and compared MLL-AF4 binding at these enhancers. We found 2550 enhancers that are bound by MLL-AF4 in at least three of the four patients, out of 3594 common enhancers (present in three or more patients). These data are included as Supplementary Fig. 2a (below).

Supplementary Figure 2. a, Overlap of enhancer usage between MLL-AF4 patients. Enhancers present in the unified MLL-AF4-bound enhancer set (bound in at least three patients) are colored purple.

6) Sentence in P2 of pg. 7 is much overstated. “From this we conclude that MLL-AF4 binding generates enhancers involved in a large-scale hub of contacts (spanning tens of kb) with target promoters to activate transcription.” This is based only on correlation of patient sample MLL-AF4 binding data with capture-C in SEM cells and does not show causation. To show that MLL-AF4 binding generates enhancers, an MLL-AF4 recruitment strategy would need to be employed

We agree with the Reviewer on this point, and we had not intended to make such a bold claim from this evidence.

We think that there is a distinction between the processes of initiating and maintaining enhancer activity. The data in this paper support a role for MLL-AF4 in maintaining enhancer activation in patients. However, it is not clear how, or whether, MLL-AF4 generates these enhancers in the first place. While many of the MLL-AF4-bound enhancers also appear to be active in normal cells (Supplementary Fig. 3f), we also find that a subset display activity at much higher levels than that seen in normal cells or other leukemias (for example see *FLT3*, Supplementary Fig. 2d, below), suggesting that these could be de novo enhancers generated by MLL-AF4 leukemia.

Supplementary Figure 2. d, ATAC-seq and H3K27ac ChIP-seq at the *FLT3* and *TNFRSF14* loci in adult blood cell types (Corces et al. 2016), MLL-AF4 patients and ALL cell lines (Kodgule et al. 2023), MLL-AF4 patients. Primary translocations are indicated. The enhancer within *PAN3* in MLL-AF4 ALL cells is highlighted in blue.

However, this doesn't address whether MLL-AF4 itself can initiate enhancer function. While there are inducible mouse models for MLL-AF9 which could be used to address this question, there is currently no inducible MLL-AF4 model. This is partly because it has been notoriously difficult to create an MLL-AF4 mouse model that recapitulates the human disease. The most recent successful model developed by us and the Roy lab uses CRISPR/Cas9 breakage of the endogenous genes (Rice et al. 2021). Two different other models use either viral expression of MLL-Af4 (a human-mouse hybrid) in rare cell types or a translocation model with or without additional miRNA "helper" expression (Lin et al. 2016, Malouf et al. 2021). None of these systems are easy to use with inducibility, for a direct "before and after" analysis of MLL-AF4 binding.

Recently, the lab of Rolf Marschalek created an inducible MLL-AF4 cell line model in 293 cells (Wilhelm and Marschalek 2021), which allows a comparison of chromatin before and after expression of MLL-AF4, albeit in a non-hematological context. We therefore analyzed their ATAC-seq data to test the possibility that this inducible model can create new active enhancers. Indeed, we found that MLL-AF4 expression was associated with increased ATAC-seq signal at non-promoter regions that might be considered enhancers (see Reviewer Figure 1a and b, below). To test whether these novel ATAC-seq peaks may be directly associated with MLL-AF4 binding, we then expressed MLL-FLAG-AF4 in 293 cells to conduct anti-FLAG ChIP-seq. We found that MLL-AF4 binding overlapped with many of these new "enhancer" sites (see Reviewer Figure 1c), but in our hands MLL-AF4 expression was toxic to 293 cells, killing them after a short period, making ATAC-seq and ChIP-seq signal very poor quality. Because of this, we weren't confident to draw conclusions from these data. We are not sure why the cells in the Marschalek study survived while ours died, but it is also arguable that 293 cells are not the correct cell type for this kind of analysis. We present our analysis here for the Reviewer, but as we were unable to repeat the Marschalek work ourselves, we are unwilling to present them in the paper.

Instead, we think that determining a role for MLL-AF4 in initiating enhancer activity requires a more thorough assessment, and will be the subject of future work. Hence, we have rewritten the paper to make it clear that we only address whether MLL-AF4 is important for the maintenance of enhancer activity in leukemia patients, and have edited the sentence referred to by the Reviewer as follows:

“[W]e conclude that, at a subset of highly active enhancers, MLL-AF4 binding is associated with a large-scale hub of contacts (spanning tens of kb) with target promoters to activate transcription.”

Reviewer Figure 1.

a, Differential accessibility analysis of 293 cells transfected with MLL-AF4 from published ATAC-seq data (Wilhelm and Marschalek 2021). Select differential ATAC-seq peaks are annotated with genomic location and nearest gene promoter. **b**, Heatmap showing the normalized accessibility of differential ATAC-seq peaks in mock transfected and MLL-AF4-expressing 293 cells. **c**, MLL-FLAG-AF4 ChIP-seq signal at differential enhancers from (b), ranked by signal intensity. Metaplot shows mean MLL-FLAG-AF4 signal at ATAC-seq peaks with increased (blue) or decreased (green) accessibility in MLL-AF4-expressing cells.

7) Fig. 4C axis labels missing (scale)

We'd like to thank the Reviewer for pointing this out; this has been fixed.

References

- Agraz-Doblas, A., C. Bueno, R. Bashford-Rogers, A. Roy, P. Schneider, M. Bardini, P. Ballerini, G. Cazzaniga, T. Moreno, C. Revilla, M. Gut, M. G. Valsecchi, I. Roberts, R. Pieters, P. De Lorenzo, I. Varela, P. Menendez and R. W. Stam (2019). "Unraveling the cellular origin and clinical prognostic markers of infant B-cell acute lymphoblastic leukemia using genome-wide analysis." *Haematologica* **104**(6): 1176-1188.
- Armstrong, S. A., J. E. Staunton, L. B. Silverman, R. Pieters, M. L. den Boer, M. D. Minden, S. E. Sallan, E. S. Lander, T. R. Golub and S. J. Korsmeyer (2002). "MLL translocations specify a distinct gene expression profile that distinguishes a unique leukemia." *Nat Genet* **30**(1): 41-47.
- Ashfield, R., A. J. Patel, S. A. Bossone, H. Brown, R. D. Campbell, K. B. Marcu and N. J. Proudfoot (1994). "MAZ-dependent termination between closely spaced human complement genes." *EMBO J* **13**(23): 5656-5667.
- Bai, X., J. Kim, Z. Yang, M. J. Jurynek, T. E. Akie, J. Lee, J. LeBlanc, A. Sessa, H. Jiang, A. DiBiase, Y. Zhou, D. J. Grunwald, S. Lin, A. B. Cantor, S. H. Orkin and L. I. Zon (2010). "TIF1gamma controls erythroid cell fate by regulating transcription elongation." *Cell* **142**(1): 133-143.
- Blackledge, N. P., A. M. Farcas, T. Kondo, H. W. King, J. F. McGouran, L. L. P. Hanssen, S. Ito, S. Cooper, K. Kondo, Y. Koseki, T. Ishikura, H. K. Long, T. W. Sheahan, N. Brockdorff, B. M. Kessler, H. Koseki and R. J. Klose (2014). "Variant PRC1 complex-dependent H2A ubiquitylation drives PRC2 recruitment and polycomb domain formation." *Cell* **157**(6): 1445-1459.
- Chen, F. X., A. R. Woodfin, A. Gardini, R. A. Rickels, S. A. Marshall, E. R. Smith, R. Shiekhattar and A. Shilatifard (2015). "PAF1, a Molecular Regulator of Promoter-Proximal Pausing by RNA Polymerase II." *Cell* **162**(5): 1003-1015.
- Chen, F. X., P. Xie, C. K. Collings, K. Cao, Y. Aoi, S. A. Marshall, E. J. Rendleman, M. Ugarenko, P. A. Ozark, A. Zhang, R. Shiekhattar, E. R. Smith, M. Q. Zhang and A. Shilatifard (2017). "PAF1 regulation of promoter-proximal pause release via enhancer activation." *Science* **357**(6357): 1294-1298.
- Corces, M. R., J. D. Buenrostro, B. Wu, P. G. Greenside, S. M. Chan, J. L. Koenig, M. P. Snyder, J. K. Pritchard, A. Kundaje, W. J. Greenleaf, R. Majeti and H. Y. Chang (2016). "Lineage-specific and single-cell chromatin accessibility charts human hematopoiesis and leukemia evolution." *Nat Genet* **48**(10): 1193-1203.
- Davies, J. O., J. M. Telenius, S. J. McGowan, N. A. Roberts, S. Taylor, D. R. Higgs and J. R. Hughes (2016). "Multiplexed analysis of chromosome conformation at vastly improved sensitivity." *Nat Methods* **13**(1): 74-80.
- Ding, L., M. Paszkowski-Rogacz, J. Mircetic, D. Chakraborty and F. Buchholz (2021). "The Paf1 complex positively regulates enhancer activity in mouse embryonic stem cells." *Life Sci Alliance* **4**(3).
- Farnung, L., M. Ochmann, G. Garg, S. M. Vos and P. Cramer (2022). "Structure of a backtracked hexasomal intermediate of nucleosome transcription." *Mol Cell* **82**(17): 3126-3134 e3127.

Figuroa, M. E., S. C. Chen, A. K. Andersson, L. A. Phillips, Y. Li, J. Sotzen, M. Kundu, J. R. Downing, A. Melnick and C. G. Mullighan (2013). "Integrated genetic and epigenetic analysis of childhood acute lymphoblastic leukemia." J Clin Invest **123**(7): 3099-3111.

Francette, A. M., S. A. Tripplehorn and K. M. Arndt (2021). "The Paf1 Complex: A Keystone of Nuclear Regulation Operating at the Interface of Transcription and Chromatin." J Mol Biol **433**(14): 166979.

Geng, H., S. Brennan, T. A. Milne, W. Y. Chen, Y. Li, C. Hurtz, S. M. Kweon, L. Zickl, S. Shojaee, D. Neuberger, C. Huang, D. Biswas, Y. Xin, J. Racevskis, R. P. Ketterling, S. M. Luger, H. Lazarus, M. S. Tallman, J. M. Rowe, M. R. Litzow, M. L. Guzman, C. D. Allis, R. G. Roeder, M. Muschen, E. Paietta, O. Elemento and A. M. Melnick (2012). "Integrative Epigenomic Analysis Identifies Biomarkers and Therapeutic Targets in Adult B-Acute Lymphoblastic Leukemia." Cancer Discov **2**(11): 1004-1023.

Godfrey, L., N. T. Crump, S. O'Byrne, I. J. Lau, S. Rice, J. R. Harman, T. Jackson, N. Elliott, G. Buck, C. Connor, R. Thorne, D. Knapp, O. Heidenreich, P. Vyas, P. Menendez, S. Inglott, P. Ancliff, H. Geng, I. Roberts, A. Roy and T. A. Milne (2021). "H3K79me2/3 controls enhancer-promoter interactions and activation of the pan-cancer stem cell marker PROM1/CD133 in MLL-AF4 leukemia cells." Leukemia **35**(1): 90-106.

Godfrey, L., N. T. Crump, R. Thorne, I. J. Lau, E. Repapi, D. Dimou, A. L. Smith, J. R. Harman, J. M. Telenius, A. M. Oudelaar, D. J. Downes, P. Vyas, J. R. Hughes and T. A. Milne (2019). "DOT1L inhibition reveals a distinct subset of enhancers dependent on H3K79 methylation." Nat Commun **10**(1): 2803.

Godfrey, L., J. Kerry, R. Thorne, E. Repapi, J. O. Davies, M. Tapia, E. Ballabio, J. R. Hughes, H. Geng, M. Konopleva and T. A. Milne (2017). "MLL-AF4 binds directly to a BCL-2 specific enhancer and modulates H3K27 acetylation." Exp Hematol **47**: 64-75.

Gustafsson, C., A. De Paepe, C. Schmidl and R. Mansson (2019). "High-throughput ChIPmentation: freely scalable, single day ChIPseq data generation from very low cell-numbers." BMC Genomics **20**(1): 59.

Harman, J. R., R. Thorne, M. Jamilly, M. Tapia, N. T. Crump, S. Rice, R. Beveridge, E. Morrissey, M. de Bruijn, I. Roberts, A. Roy, T. A. Fulga and T. Milne (2021). "A KMT2A-AFF1 gene regulatory network highlights the role of core transcription factors and reveals the regulatory logic of key downstream target genes." Genome Res.

Harvey, R. C., C. G. Mullighan, X. Wang, K. K. Dobbin, G. S. Davidson, E. J. Bedrick, I. M. Chen, S. R. Atlas, H. Kang, K. Ar, C. S. Wilson, W. Wharton, M. Murphy, M. Devidas, A. J. Carroll, M. J. Borowitz, W. P. Bowman, J. R. Downing, M. Relling, J. Yang, D. Bhojwani, W. L. Carroll, B. Camitta, G. H. Reaman, M. Smith, S. P. Hunger and C. L. Willman (2010). "Identification of novel cluster groups in pediatric high-risk B-precursor acute lymphoblastic leukemia with gene expression profiling: correlation with genome-wide DNA copy number alterations, clinical characteristics, and outcome." Blood **116**(23): 4874-4884.

Heinz, S., C. Benner, N. Spann, E. Bertolino, Y. C. Lin, P. Laslo, J. X. Cheng, C. Murre, H. Singh and C. K. Glass (2010). "Simple combinations of lineage-determining transcription factors prime cis-regulatory elements required for macrophage and B cell identities." Mol Cell **38**(4): 576-589.

Hentges, L. D., M. J. Sergeant, D. J. Downes, J. R. Hughes and S. Taylor (2021). "LanceOtron: a deep learning peak caller for ATAC-seq, ChIP-seq, and DNase-seq." bioRxiv: 2021.2001.2025.428108.

- Hou, L., Y. Wang, Y. Liu, N. Zhang, I. Shamovsky, E. Nudler, B. Tian and B. D. Dynlacht (2019). "Paf1C regulates RNA polymerase II progression by modulating elongation rate." Proc Natl Acad Sci U S A **116**(29): 14583-14592.
- Hsieh, T. H., A. Weiner, B. Lajoie, J. Dekker, N. Friedman and O. J. Rando (2015). "Mapping Nucleosome Resolution Chromosome Folding in Yeast by Micro-C." Cell **162**(1): 108-119.
- Hua, P., M. Badat, L. L. P. Hanssen, L. D. Hentges, N. Crump, D. J. Downes, D. M. Jeziorska, A. M. Oudelaar, R. Schwessinger, S. Taylor, T. A. Milne, J. R. Hughes, D. R. Higgs and J. O. J. Davies (2021). "Defining genome architecture at base-pair resolution." Nature **595**(7865): 125-129.
- Kent, W. J., C. W. Sugnet, T. S. Furey, K. M. Roskin, T. H. Pringle, A. M. Zahler and D. Haussler (2002). "The human genome browser at UCSC." Genome Res **12**(6): 996-1006.
- Kerry, J., L. Godfrey, E. Repapi, M. Tapia, N. P. Blackledge, H. Ma, E. Ballabio, S. O'Byrne, F. Ponthan, O. Heidenreich, A. Roy, I. Roberts, M. Konopleva, R. J. Klose, H. Geng and T. A. Milne (2017). "MLL-AF4 Spreading Identifies Binding Sites that Are Distinct from Super-Enhancers and that Govern Sensitivity to DOT1L Inhibition in Leukemia." Cell Rep **18**(2): 482-495.
- Kim, J., M. Guermah and R. G. Roeder (2010). "The human PAF1 complex acts in chromatin transcription elongation both independently and cooperatively with SII/TFIIS." Cell **140**(4): 491-503.
- Kodgule, R., J. W. Goldman, A. C. Monovich, T. Saari, A. R. Aguilar, C. N. Hall, N. Rajesh, J. Gupta, S. A. Chu, L. Ye, A. Gurumurthy, A. Iyer, N. A. Brown, M. Y. Chiang, M. P. Cieslik and R. J. H. Ryan (2023). "ETV6 Deficiency Unlocks ERG-Dependent Microsatellite Enhancers to Drive Aberrant Gene Activation in B-Lymphoblastic Leukemia." Blood Cancer Discov **4**(1): 34-53.
- Langmead, B., C. Trapnell, M. Pop and S. L. Salzberg (2009). "Ultrafast and memory-efficient alignment of short DNA sequences to the human genome." Genome Biol **10**(3): R25.
- Li, H., B. Handsaker, A. Wysoker, T. Fennell, J. Ruan, N. Homer, G. Marth, G. Abecasis, R. Durbin and S. Genome Project Data Processing (2009). "The Sequence Alignment/Map format and SAMtools." Bioinformatics **25**(16): 2078-2079.
- Lin, S., R. T. Luo, A. Ptasinska, J. Kerry, S. A. Assi, M. Wunderlich, T. Imamura, J. J. Kaberlein, A. Rayes, M. J. Althoff, J. Anastasi, M. M. O'Brien, A. R. Meetei, T. A. Milne, C. Bonifer, J. C. Mulloy and M. J. Thirman (2016). "Instructive Role of MLL-Fusion Proteins Revealed by a Model of t(4;11) Pro-B Acute Lymphoblastic Leukemia." Cancer Cell **30**(5): 737-749.
- Liu, X., Z. Guo, J. Han, B. Peng, B. Zhang, H. Li, X. Hu, C. J. David and M. Chen (2022). "The PAF1 complex promotes 3' processing of pervasive transcripts." Cell Rep **38**(11): 110519.
- Lu, X., X. Zhu, Y. Li, M. Liu, B. Yu, Y. Wang, M. Rao, H. Yang, K. Zhou, Y. Wang, Y. Chen, M. Chen, S. Zhuang, L. F. Chen, R. Liu and R. Chen (2016). "Multiple P-TEFbs cooperatively regulate the release of promoter-proximally paused RNA polymerase II." Nucleic Acids Res **44**(14): 6853-6867.

Malouf, C., E. T. B. Antunes, M. O'Dwyer, H. Jakobczyk, F. Sahm, S. L. Landua, R. A. Anderson, A. Soufi, C. Halsey and K. Ottersbach (2021). "miR-130b and miR-128a are essential lineage-specific codrivers of t(4;11) MLL-AF4 acute leukemia." Blood **138**(21): 2066-2092.

O'Byrne, S., N. Elliott, S. Rice, G. Buck, N. Fordham, C. Garnett, L. Godfrey, N. T. Crump, G. Wright, S. Inglott, P. Hua, B. Psaila, B. Povinelli, D. Knapp, A. Agraz-Doblas, C. Bueno, I. Varela, P. Bennett, H. Koohy, S. M. Watt, A. Karadimitris, A. J. Mead, P. Ancliff, P. Vyas, P. Menendez, T. A. Milne, I. Roberts and A. Roy (2019). "Discovery of a CD10-negative B-progenitor in human fetal life identifies unique ontogeny-related developmental programs." Blood **134**(13): 1059-1071.

Prange, K. H. M., A. Mandoli, T. Kuznetsova, S. Y. Wang, A. M. Sotoca, A. E. Marneth, B. A. van der Reijden, H. G. Stunnenberg and J. H. A. Martens (2017). "MLL-AF9 and MLL-AF4 oncofusion proteins bind a distinct enhancer repertoire and target the RUNX1 program in 11q23 acute myeloid leukemia." Oncogene **36**(23): 3346-3356.

Rada-Iglesias, A., R. Bajpai, T. Swigut, S. A. Brugmann, R. A. Flynn and J. Wysocka (2011). "A unique chromatin signature uncovers early developmental enhancers in humans." Nature **470**(7333): 279-283.

Ramirez, F., D. P. Ryan, B. Gruning, V. Bhardwaj, F. Kilpert, A. S. Richter, S. Heyne, F. Dunder and T. Manke (2016). "deepTools2: a next generation web server for deep-sequencing data analysis." Nucleic Acids Res **44**(W1): W160-165.

Rice, S., T. Jackson, N. T. Crump, N. Fordham, N. Elliott, S. O'Byrne, M. Fanego, D. Addy, T. Crabb, C. Dryden, S. Inglott, D. Ladon, G. Wright, J. Bartram, P. Ancliff, A. J. Mead, C. Halsey, I. Roberts, T. A. Milne and A. Roy (2021). "A human fetal liver-derived infant MLL-AF4 acute lymphoblastic leukemia model reveals a distinct fetal gene expression program." Nat Commun **12**(1): 6905.

Ross, M. E., X. Zhou, G. Song, S. A. Shurtleff, K. Girtman, W. K. Williams, H. C. Liu, R. Mahfouz, S. C. Raimondi, N. Lenny, A. Patel and J. R. Downing (2003). "Classification of pediatric acute lymphoblastic leukemia by gene expression profiling." Blood **102**(8): 2951-2959.

Van Oss, S. B., C. E. Cucinotta and K. M. Arndt (2017). "Emerging Insights into the Roles of the Paf1 Complex in Gene Regulation." Trends Biochem Sci **42**(10): 788-798.

Vos, S. M., L. Farnung, M. Boehning, C. Wigge, A. Linden, H. Urlaub and P. Cramer (2018). "Structure of activated transcription complex Pol II-DSIF-PAF-SPT6." Nature **560**(7720): 607-612.

Wilhelm, A. and R. Marschalek (2021). "The role of reciprocal fusions in MLL-r acute leukemia: studying the chromosomal translocation t(4;11)." Oncogene **40**(42): 6093-6102.

Wilkinson, A. C., E. Ballabio, H. Geng, P. North, M. Tapia, J. Kerry, D. Biswas, R. G. Roeder, C. D. Allis, A. Melnick, M. F. de Bruijn and T. A. Milne (2013). "RUNX1 is a key target in t(4;11) leukemias that contributes to gene activation through an AF4-MLL complex interaction." Cell Rep **3**(1): 116-127.

Wu, L., L. Li, B. Zhou, Z. Qin and Y. Dou (2014). "H2B ubiquitylation promotes RNA Pol II processivity via PAF1 and pTEFb." Mol Cell **54**(6): 920-931.

Yu, M., W. Yang, T. Ni, Z. Tang, T. Nakadai, J. Zhu and R. G. Roeder (2015). "RNA polymerase II-associated factor 1 regulates the release and phosphorylation of paused RNA polymerase II." Science **350**(6266): 1383-1386.

Zumer, K., K. C. Maier, L. Farnung, M. G. Jaeger, P. Rus, G. Winter and P. Cramer (2021). "Two distinct mechanisms of RNA polymerase II elongation stimulation in vivo." Mol Cell **81**(15): 3096-3109 e3098.

Reviewers' Comments:

Reviewer #1:

Remarks to the Author:

The authors have responded to all of my concerns in a satisfactory way, and I have no more comments to add.

Reviewer #2:

Remarks to the Author:

The authors have satisfactorily addressed all the reviewers' comments. I have no additional points to raise.

Reviewer #3:

Remarks to the Author:

The authors have thoroughly addressed all concerns. The additional experimental data and analyses support that the MLL::AF4 FP plays an important role in maintaining a hyper-active enhancer state at a subset of enhancers in leukemia cells. There is likely to be a critical role for enhancer RNA transcription in the maintenance of this hyper-active enhancer state, hence the requirement for PAF1/FACT at these loci. How MLL::AF4 specifically targets these sites remains an open question, as well as the interesting observation of global H3K27ac loss following MLL::AF4 loss

Response to reviewers

We would like to thank all three reviewers for having taken the time to review our paper, and we are gratified that we have satisfied their concerns with the work. We agree with Reviewer #3 that these additional experiments and analyses have strengthened the paper, and opens up exciting avenues for future research.

Reviewer #1

The authors have responded to all of my concerns in a satisfactory way, and I have no more comments to add.

Reviewer #2

The authors have satisfactorily addressed all the reviewers' comments. I have no additional points to raise.

Reviewer #3

The authors have thoroughly addressed all concerns. The additional experimental data and analyses support that the MLL::AF4 FP plays an important role in maintaining a hyper-active enhancer state at a subset of enhancers in leukemia cells. There is likely to be a critical role for enhancer RNA transcription in the maintenance of this hyper-active enhancer state, hence the requirement for PAF1/FACT at these loci. How MLL::AF4 specifically targets these sites remains an open question, as well as the interesting observation of global H3K27ac loss following MLL::AF4 loss